# Near-resonant nuclear spin detection with megahertz mechanical resonators

Diego A. Visani[1,2], Letizia Catalini[1,2,3], Christian L. Degen[1,2], Alexander Eichler[1,2,*], Javier del Pino[4,5]

[1]Laboratory for Solid State Physics, ETH Zürich, 8093 Zürich, Switzerland
[2]Quantum Center, ETH Zürich, 8093 Zürich, Switzerland
[3]Center for Nanophotonics, AMOLF, 1098XG Amsterdam, The Netherlands
[4]Institute for Theoretical Physics, ETH Zürich, 8093 Zürich, Switzerland
[5]Departamento de Física Teórica de la Materia Condensada and Condensed Matter Physics
Center (IFIMAC), Universidad Autónoma de Madrid, E28049 Madrid, Spain

*eichlera@ethz.ch

## Abstract

**Mechanical resonators operating in the megahertz range have become a versatile platform for fundamental and applied quantum research. Their exceptional properties, such as low mass and high quality factor, make them also appealing for force sensing experiments. In this work, we propose a method for detecting, and ultimately controlling, nuclear spins by coupling them to megahertz resonators via a magnetic field gradient. Dynamical backaction between the sensor and an ensemble of $N$ nuclear spins produces a shift in the sensor's resonance frequency. The mean frequency shift due to the Boltzmann polarization is challenging to measure in nanoscale sample volumes. Here, we show that the fluctuating polarization of the spin ensemble results in a measurable increase of the resonator's frequency variance. On the basis of analytical as well as numerical results, we predict that the variance measurement will allow single nuclear spin detection with existing resonator devices.**

# 1  Introduction

Magnetic resonance force microscopy (MRFM) is a method to achieve nanoscale magnetic resonance imaging (MRI) [1,2]. It relies on a mechanical sensor interacting via a magnetic field gradient with an ensemble of nuclear spins. The interaction creates signatures in the resonator oscillation that can be used to detect nuclear spins with high spatial resolution. Previous milestones include the imaging of virus particles with $5 - 10\,\mathrm{nm}$ resolution [3], Fourier-transform nanoscale MRI [4], nuclear spin detection with a one-dimensional resolution below $1\,\mathrm{nm}$ [5], and magnetic resonance diffraction with subangstrom precision [6].

The MRFM community is continuously searching for improved force sensors to reach new regimes of spin-mechanics interaction. In particular, over the last decade, new classes of mechanical resonators made from strained materials showed promise as force sensors [7]. Today, these resonators come in a large variety of designs, including trampolines [8, 9], membranes [10–12], strings [13–16], polygons [17], hierarchical structures [18, 19], and spider webs [20]. Some of these resonators are massive enough to be seen by the naked eye, but their low dissipation nevertheless makes them excellent sensors, potentially on par with carbon nanotubes [21] and nanowires [4, 22].

Compared to the cantilevers and nanowires traditionally used in MRFM, the new classes of mechanical resonators typically exhibit higher resonance frequencies and different shapes. As a consequence, protocols used in previous MRFM experiments are often not applicable anymore. On the one hand, this calls for novel scanning force geometries [23] and transduction protocols [24] that are tailored to the new sensors. On the other hand, new experimental opportunities arise, as these mechanical resonators can strongly interact with a wide array of quantum systems, such as nuclear spins, artificial atoms, and photonic resonators [25, 26].

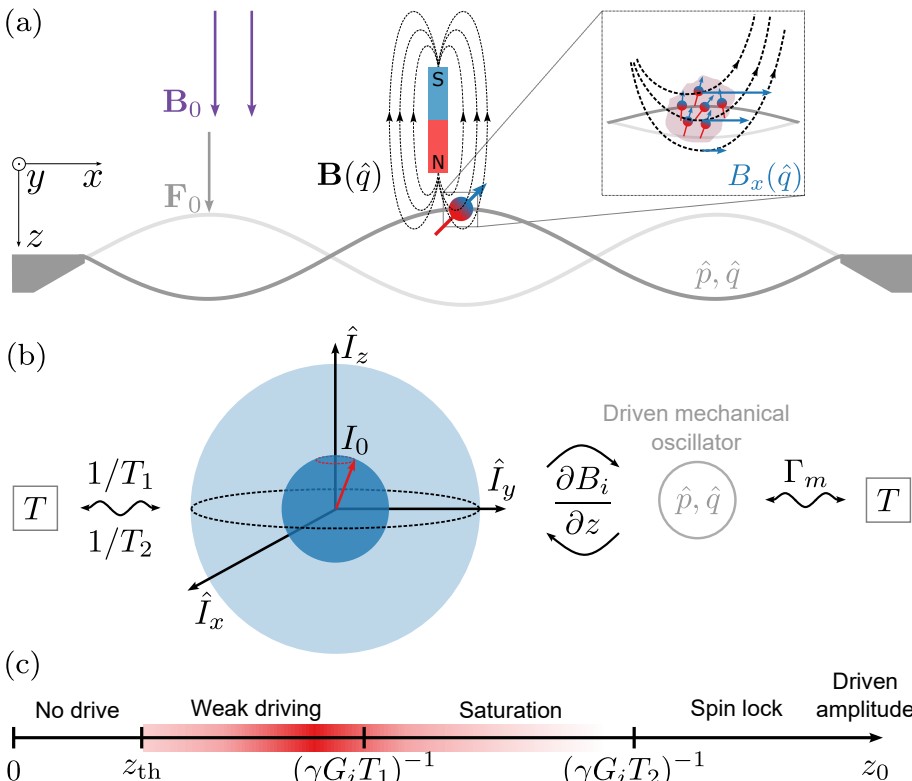

Figure 1: (a) Schematic of the proposed experiment: A spin ensemble is placed on a mechanical resonator moving within an inhomogeneous magnetic field generated by a nanoscale magnet. By driving the resonator, the spins experience an oscillating magnetic field $\mathbf{B}(\hat{q})$ with a component $B_x$ (inset). The spins act back on the resonator, producing a force that can be detected as a shift in the resonance frequency. (b) We model the system as a spin ensemble (equilibrium polarization $I_\parallel = I_0$) interacting with a harmonic oscillator. Both spin ensemble and resonator are coupled to independent baths at temperature $T$, causing spin dephasing and decay with rates $1/T_2$ and $1/T_1$, respectively, and resonator damping at a rate $\Gamma_{\mathrm{m}}$. (c) Illustration of the typical spin regimes according to driven mechanical amplitudes $z_0$. Here $z_{\mathrm{th}}$ denotes the thermal motional amplitude. The regime addressed in this work is highlighted in red.

In this work, we propose a protocol for nuclear spin detection based on the near-resonant interaction between a mechanical resonator and nuclear spins. We start from a general case with a large ensemble of nuclear spins and develop a deterministic model of the near-resonant interaction. We then extend this framework to small sample volumes, where statistical effects dominate, down to the limit of a single fluctuating nuclear spin. Opposed to earlier ideas [27,28], our method is most efficient when the resonator is slightly detuned from the spin Larmor frequency. Our method suits the typical frequency range of strained silicon nitride resonators ($1 - 50\,\mathrm{MHz}$) and offers a simplified experimental apparatus, as it circumvents the need for spin inversion pulses and related hardware. We also show that for realistic experimental parameters, the method can attain single nuclear spin sensitivity, a major milestone on the way towards spin-based quantum devices. Finally, our method will enable spin manipulation via mechanical driving, in analogy to existing techniques in cavity optomechanics [25, 29–31].

## 2 Theoretical Framework

We first consider a nuclear spin ensemble placed on a mechanical resonator, see Fig. 1(a). The ensemble comprises $N$ spins that interact with a normal mode of the resonator. The composite system can be described with the Zeeman-like Hamiltonian

$$\mathcal{H} = -\hbar\gamma\hat{\mathbf{I}} \cdot \mathbf{B} + \mathcal{H}_m, \tag{1}$$

where $\hbar$ is the reduced Planck constant, $\gamma$ the nuclear spins' gyromagnetic ratio, and $\mathbf{B}$ the magnetic field at the spins' location. The spin ensemble operator $\hat{\mathbf{I}}$ has the three components $\hat{I}_i = \sum_{k=1}^{N} \hat{\sigma}_{i,k}/2$ with the spin-$\frac{1}{2}$ Pauli matrices $\hat{\sigma}_{i,k}$ for spin $k = \{1, \cdots, N\}$, and $i \in [x, y, z]$. We describe a single vibrational mode as a driven harmonic oscillator displacing along the $z$ axis governed by the Hamiltonian

$$\mathcal{H}_m = \frac{\hat{p}^2}{2m} + \frac{1}{2}m\omega_0^2\hat{q}^2 - F_0\hat{q}\cos(\omega_{\mathrm{d}}t), \tag{2}$$

where $\hat{q}$ is the $z$-position operator of the resonator, $\hat{p}$ is the corresponding momentum operator, $m$ is the effective mass, $\omega_0$ is the angular resonance frequency, and $\omega_{\mathrm{d}}$ and $F_0$ are the angular frequency and strength of an applied force, respectively. If $\mathbf{B}$ is inhomogeneous, the spins experience a position-dependent field $\mathbf{B}(\hat{q})$ as the mechanical resonator vibrates. To lowest order, we approximate this field as $\mathbf{B}(\hat{q}) \approx \mathbf{B}_0 + \mathbf{G}\hat{q}$ with a constant component $\mathbf{B}_0 = \mathbf{B}(\hat{q} = 0)$ and relevant field gradients $G_i = \partial B_i/\partial z$. The coherent spin-resonator dynamics therefore obey the Hamiltonian

$$\mathcal{H} \approx -\hbar\omega_{\mathrm{L}}\hat{I}_z - \hbar\gamma\hat{q}\mathbf{G} \cdot \hat{\mathbf{I}} + \mathcal{H}_m, \tag{3}$$

with the Larmor precession frequency $\omega_{\mathrm{L}} = \gamma|B_0|$.

Any real system, in equilibrium with a thermal bath, experiences mechanical damping (rate $\Gamma_{\mathrm{m}} = \omega_0/Q$, with $Q$ the quality factor), spin decay (longitudinal relaxation time $T_1$), and spin decoherence (transverse relaxation time $T_2$). We thus succinctly represent our system dynamics using the Heisenberg picture's dissipative equations of motion (EOM). Driving the resonator to an oscillation amplitude $z_0$ well above its zero-point fluctuation amplitude $z_{\mathrm{zpf}} = \sqrt{\hbar/(2m\omega_0)}$, the mechanical resonator behaves essentially classically. This allows us to assume the semiclassical limit for spins $\hat{I}_i \mapsto I_i$. The spin components $I_i$ evolve according to [Appendix A.2]

$$\ddot{q} = -\omega_0^2 q - \Gamma_{\mathrm{m}}\dot{q} + \frac{F_0}{m}\cos(\omega_{\mathrm{d}}t) + \frac{\hbar\gamma}{m}\mathbf{G} \cdot \mathbf{I} + \xi(t), \tag{4}$$

$$\dot{I}_{x,y} = -\frac{1}{T_2}I_{x,y} \pm (\omega_{\mathrm{L}} + \gamma q G_z)I_{y,x} \mp \gamma q G_{y,x}I_z, \tag{5}$$

$$\dot{I}_z = \frac{1}{T_1}\left(\zeta_0(t) - I_z\right) - \gamma q\left(G_x I_y - G_y I_x\right). \tag{6}$$

Here, the stochastic driving term $\xi(t)$ represents the thermomechanical (white) force noise. The term $\zeta_0(t) = I_0 + \delta I_0(t)$ contains two contributions: (i) the Boltzmann polarization $I_0$, representing the net equilibrium polarization of the spin ensemble. It arises due to the thermal population imbalance between spin states in the presence of an external magnetic field. In the limit $k_B T \gg \hbar\omega_{\mathrm{L}}$, the Boltzmann polarization simplifies $I_0 \approx N\hbar\omega_{\mathrm{L}}/(4k_B T)$ according to the Curie law [32]. (ii) The fluctuating statistical part $\delta I_0$ arises from thermal fluctuations in the spin ensemble. The central limit theorem dictates that these fluctuations have zero mean and a standard deviation $\sigma_{\delta I_0} \approx \sqrt{N}/2$, independent of temperature and magnetic field, in the same limit $k_B T \gg \hbar\omega_{\mathrm{L}}$ [33, 34].

To model the dynamics of these fluctuations, we assume that $\delta I_0(t)$ follows an Ornstein-Uhlenbeck process, which describes a stationary, Gaussian, Markovian process with auto-correlation $\langle \delta I_0(t) \delta I_0(t') \rangle = \sigma_{\delta I_0}^2 e^{-|t-t'|/\tau}$, decaying exponentially over the spin correlation time $\tau \leq T_1$ [35]. Note that in our model, the spins' decay (decoherence) time $T_1$ ($T_2$) is independent of temperature and magnetic field.

To treat Eqs. (4)-(6), we make a number of simplifications. In particular, we assume that: (i) the spins' force on the resonator, $\delta F$, is substantially weaker than the driving force, i.e., $|\delta F| \ll F_0$; and that (ii) the spin-resonator coupling, measured by the Rabi frequency $\Omega_R = \gamma G_i z_0$, is significantly smaller than the spin's decoherence rate, i.e., $\Omega_R \ll 1/T_2$, see Fig. 1(c). We select $z_0$ to be small and on the order of the thermal motion $z_{\text{th}}$, thus fulfilling (ii). The conditions (i) and (ii) imply that we remain in the weak coupling limit, where the oscillation inside the field gradient $\mathbf{G}$ excites a precessing spin polarization orthogonal to $\mathbf{B}_0$ (i.e., $I_{x,y} \neq 0$), but does not lock the spins to the resonator frequency $\omega_0$. The backaction of the spins can be treated as a perturbation of the driven resonator oscillation at frequency $\omega_d$. Additionally, we assume that (iii) the resonator reacts much more slowly than the spin relaxation timescales, $\Gamma_m \ll 1/T_2, 1/T_1$. Finally, we assume that (iv) spin fluctuations evolve on timescales comparable to or slower than the resonator response, i.e., $\Gamma_m \gg 1/\tau$. This ensures that spin noise can be effectively sampled by the resonator.

Since individual spins relax on a timescale set by $T_1$, the correlation time $\tau$ cannot exceed $T_1$, as any collective memory in the spin bath is lost beyond that point—an upper bound that may seem at odds with (iii), which requires $\Gamma_m \ll 1/T_1$; while both conditions cannot strictly hold simultaneously, our numerical simulations show that the analytical treatment remains valid for a wide range of spin-resonator couplings, longitudinal relaxation timescales and correlation times, including cases where $\Gamma_m \sim 1/T_1$, $\Omega_R \sim 1/T_1$ and $\tau \sim T_1$. All conditions above ensure that the response remains linear in both $I_0$ and $\delta I_0$, requiring them to be weak enough for the resonator to stay in the linear regime. For numerical validation and more information, see Appendix C.

## 3  Boltzmann Polarization

In a first part, we ignore spin fluctuations ($\delta I_0 = 0$). The spin components $I_{x,y}$ exert a linear force onto the resonator, which we calculate via the Harmonic Balance method [36, 37], detailed in Appendix A.2.1. The force involves a static component $\delta F_0 = \hbar \gamma I_0 G_z / m$ that shifts the mechanical equilibrium position, and two oscillating components, one in phase and one out of phase. This *dynamical backaction* loop causes a frequency shift $\delta \omega$ (corresponding to a phase shift in the driven response) and a linewidth change $\delta \Gamma_m$

$$\delta\omega = -g^2 \left( \frac{\omega_+}{\omega_+^2 + T_2^{-2}} + \frac{\omega_-}{\omega_-^2 + T_2^{-2}} \right) I_0, \tag{7}$$

$$\delta\Gamma_m = -g^2 \left( \frac{T_2^{-1}}{\omega_+^2 + T_2^{-2}} - \frac{T_2^{-1}}{\omega_-^2 + T_2^{-2}} \right) I_0, \tag{8}$$

where $\omega_\pm = \omega_L \pm \omega_d$ and $g^2 = \hbar \gamma^2 \left( G_x^2 + G_y^2 \right) / (4 m \omega_d)$. Equations (7) and (8) show that the average spin polarization $I_0$ affects the resonator response. The in-plane spin components $I_x$ and $I_y$ produce a delayed force. In the numerical simulation below, we find that this delayed force is strongest when the spins respond faster than the resonator, i.e., $\Gamma_m \ll 1/T_1$ and typically also $\Gamma_m \ll 1/T_2$. This is in agreement with condition (iii).

In Eq. (7) the largest frequency shift occurs at a detuning $\omega_L \neq \omega_0$, set by $1/T_2$ and $\omega_L$. This contrasts with resonant coupling forces ($\omega_L = \omega_0$) in an early MRFM proposal [27] and with spin noise measurements in MRI [38,39]. Instead, the effect resembles dynamical back-action in cavity optomechanics, where mechanical motion induces a periodic shift in an optical cavity, resulting in a corresponding change in the cavity population [25,40]. Note that, unlike spin systems fluctuating around a strongly polarized $z$ state, which involve only two oscillating quadratures [41,42], our system engages all three spin components in the back-action loop.

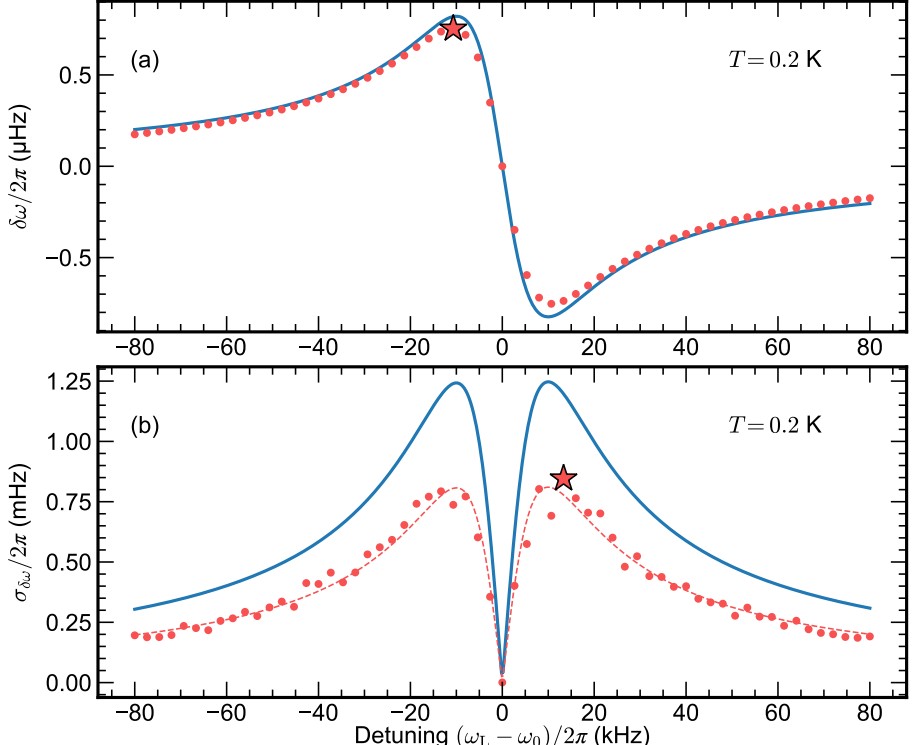

Figure 2: Mean ($\delta\omega$) and standard deviation ($\sigma_{\delta\omega}$) of the frequency shift of a string resonator [13] due to a single proton spin calculated as a function of the detuning between the Larmor frequency $\omega_L$ and the mechanical frequency $\omega_0$. Analytical and numerical results are shown for the two contributions of the spin polarization: Boltzmann (a) and statistical (b). Note the different y-axis scales. Blue lines correspond to Eq. (7), while red dots are calculated with an explicit Runge-Kutta method of order 8 [43]. Maximal frequency shifts and variances are marked with red stars. In (b), a fit of the data is shown as a red dashed line, which is identical to the analytical solution up to a factor $\eta = 0.65$. Common simulated parameters are $\omega_d = \omega_0 = 2\pi \times 5.5\,\text{MHz}$, $G_x = G_y = 6\,\text{MT/m}$, $G_z = 1\,\text{MT/m}$, $m = 2\,\text{pg}$, $T_1 = \tau = 50\,\text{ms}$, $T_2 = 100\,\mu\text{s}$, $N = 1$ and $Q_{\text{eff}} = 2 \cdot 10^4$ (see Appendix C for details).

As an example, we consider a single nuclear spin ($N = 1$) without fluctuations ($\delta I_0 = 0$) in a magnetic field of $B_0 = 130\,\text{mT}$, interacting with a bath at $T = 0.2\,\text{K}$ and a state-of-the-art string resonator [13]. The analytical results of Eqs. (7) for Boltzmann polarization are shown in Fig. 2(a), along with a numerical simulation of Eqs. (4)-(6). The analytical and numerical results show excellent agreement, with a peak frequency shift near $10\,\text{kHz}$ detuning. However, we note two issues: on the one hand, the condition $\delta I_0 = 0$ is unrealistic for any measurement time larger than $\tau$. On the other hand, we

see that even within that short time, the frequency shift that can be obtained for a single spin is only about $0.8\,\mu\mathrm{Hz}$. Measuring such a small shift is clearly unfeasible. For both of the above reasons, it appears advantageous to investigate the effects of a stochastic polarization $\delta I_0$, which should yield a much larger signal than $I_0$ for single spins [33, 44].

# 4   Statistical polarization

It is known that the statistical polarization dominates over Boltzmann polarization for $N < 2 \cdot 10^6$ spins, corresponding to a volume of $\approx (30\,\mathrm{nm})^3$ for protons in water [3, 33], see also Appendix C.3. However, from our derivation, it is unclear whether Eqs. (7) and (8) apply to statistical polarization at all. Naively, we are tempted to just replace $I_0$ with $\delta I_0$, making $\delta\omega$ and $\delta\Gamma_\mathrm{m}$ explicitly time-dependent and stochastic. This would entail that the variance of the frequency shift reflects fluctuations in the spin bath spectral density. A detailed derivation (see Appendix A.2.2) confirms this intuition: assuming Gaussian statistics and that the conditions (i), (ii), and (iii) behind Eqs.(7) still hold—namely, weak coupling, fast dephasing, and narrow-band resonator response—we find via standard error propagation:

$$\sigma_{\delta\omega} = g^2 \left( \frac{\omega_+}{1/T_2^2 + \omega_+^2} + \frac{\omega_-}{1/T_2^2 + \omega_-^2} \right) \sigma_{\delta I_0}. \tag{9}$$

The relevant observable in this scenario is no longer a static frequency shift but the standard deviation of frequency fluctuations, $\sigma_{\delta\omega}$. As Eq. (9) shows, this standard deviation scales as $\sigma_{\delta I_0} \sim \sqrt{N}$. We find that the variance of the frequency shift peaks at the same parameter values where the average shift is largest. This is expected: both the mean shift and its fluctuations grow with the strength of the spin-resonator interaction.

In Fig. 2(b), we show the analytical result corresponding to Eqs. (9), calculated for the same resonator and a single proton spin. As before, we compare the analytical results to a numerical simulation of the semiclassical, stochastic Eqs. (4)-(6), which we now carry out for a fluctuating polarization. We simulate multiple stochastic trajectories of Eqs. (4)–(6) using a long spin correlation time $\tau = 50\,\mathrm{ms} = T_1$, compute their variance $\sigma_{\delta\omega}^\mathrm{sim}$, and remove transients from initialization. Crucially, we find that $\sigma_{\delta\omega}^\mathrm{sim}$ in this case is ca. 3 orders of magnitude larger than the frequency shift shown for the Boltzmann case in Fig. 2(a). Indeed, the standard deviation expected for a single proton approaches $1\,\mathrm{mHz}$, which should be measurable at cryogenic temperatures [45]. We conclude that measuring the statistical spin polarization is promising and could enable single nuclear spin detection.

The full numerical simulation shows that the analytical prediction in Eq. (9) overestimates $\sigma_{\delta\omega}$ by the factor $\eta \equiv \sigma_{\delta\omega}^\mathrm{sim}/\sigma_{\delta\omega}$. For the example shown in Fig. 2(b), we find $\eta = 0.65$. The discrepancy arises from two factors: on the one hand, if $\Gamma_\mathrm{m}$ is small relative to $1/\tau$, the resonator cannot sample the fluctuating spin polarization sufficiently fast. This corresponds to a violation of condition (iv). On the other hand, if $\Gamma_\mathrm{m}$ is large relative to $1/T_1$, we violate condition (iii) and the analytical result is unrealistic. As $\tau \sim T_1$, the two conditions cannot be perfectly fulfilled at the same time and we expect to always obtain an overall reduction compared to the analytical prediction.

To further investigate this reduction, we show $\eta$ as a function of $\Gamma_\mathrm{m}$ and $\tau$ in Figs. 3(a) and (b), respectively. We observe a monotonic reduction of $\eta$ for fast spin baths ($\tau \to 0$) and for high-$Q$ resonators (slow response). Appendix A.2.2 presents a detailed model of spin-force under statistical polarization and its impact on the resonator. For simplicity, we adopt here a phenomenological approach, assuming a Lorentzian spin-force PSD set by a fixed spin correlation time [24]. In this simplified model, we calculate the PSD of the fluctuating mechanical frequency from the spectral overlap of the spin force PSD with the

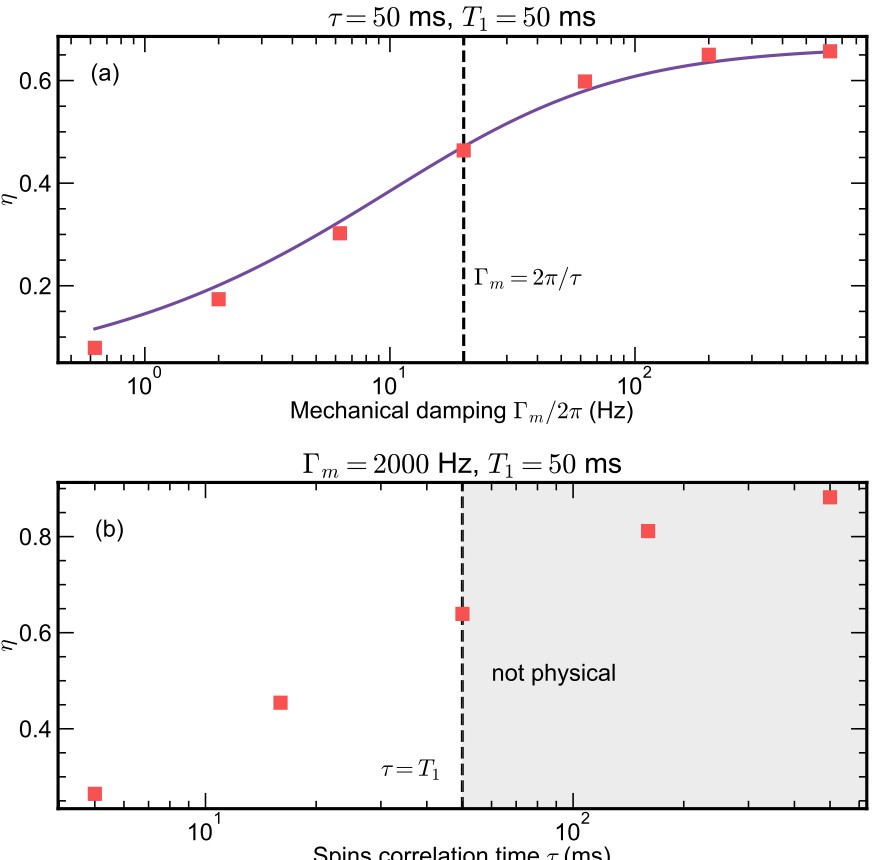

Figure 3: Dependence of simulated frequency shift variance on the resonator's damping rate $\Gamma_m$ (a) and on the spin bath correlation time $\tau$ (b), quantified by the factor $\eta$ extracted from fits to multiple numerical simulations. The dashed vertical line in (a) indicates the point where resonator's response time and spin correlation's time match. In addition, a purple line shows a single parameter fit of the model described by Eq. (11). The fit gives $\alpha = 0.67$. In (b), a dashed vertical line marks the theoretical upper limit $\tau = T_1$, beyond which the bath cannot stay correlated.

mechanical resonator response function. i.e., the mechanical susceptibility:

$$S_{\delta\omega\delta\omega}(\omega) = \frac{1}{4z_0^2 m^2 \omega_0^2} \frac{\Gamma_m^2}{\Gamma_m^2 + (\omega - \omega_0)^2} \frac{\left(\frac{2\pi}{\tau}\right)}{\left(\frac{2\pi}{\tau}\right)^2 + (\omega - \omega_L)^2} F_{spin}^2, \qquad (10)$$

where $F_{spin}$ is the force generated by the spins, assumed to be frequency-independent. Note that the exact form of the force does not need to be known for the model to properly describe the factor $\eta$. Using $\sigma_{\delta\omega}^2 = \int_{-\infty}^{\infty} \frac{d\omega}{2\pi} S_{\delta\omega\delta\omega}(\omega)$ and normalizing to $\tau \to \infty$ (i.e. the non fluctuating polarization limit), we get:

$$\eta = \alpha \frac{\Gamma_m \tau}{2\pi + \Gamma_m \tau}, \qquad (11)$$

with $\alpha$ the only fit parameter of the model that accounts for all the prefactors in Eq. (10), including the unknown form of $F_{spin}$. Note that $\alpha$ sets the maximum value of $\eta$ for a given set of $\Gamma_m$ and $\tau$.

The purple line on Fig. 3(a) shows the model with the single fit parameter $\alpha = 0.67$. We observe that larger values of $\Gamma_{\mathrm{m}}$ lead to higher $\eta$ as they conform closer to condition (iv), that is, the resonator is better able to sample the spin fluctuations in real time. Nevertheless, we do not reach $\eta = 1$. We attribute this to the partial violation of condition (iii), which arises because $\tau \leq T_1$ by necessity. In Fig. 3(b), we demonstrate how $\eta$ depends on $\tau$. Indeed, we find that for arbitrarily large $\tau$, $\eta$ converges towards 1. This regime is marked as 'not physical' as it corresponds to $\tau > T_1$.

Note that condition (iv) is not fundamental, in the sense that a resonator can sample frequency shifts much faster than its own ringdown time $\approx 1/\Gamma_{\mathrm{m}}$ when using a closed-loop measurement technique [46, 47]. For such a closed-loop technique, all points in Fig. 3(a) would have the same value of 0.67.

To estimate the smallest measurable frequency shift, we compare its variance with the resonator's frequency noise from thermal fluctuations. At resonance ($\omega = \omega_0$), the power spectral density of the resonator's displacement reads $S_{qq}(\omega_0) = 4k_B T Q/(m\omega_0^3)$. These displacement fluctuations translate into frequency noise with spectral density [47]:

$$S_{\delta\omega\delta\omega}(\omega_0) = \frac{2\omega_0^2 S_{qq}(\omega_0)}{4Q^2 z_0^2} = \frac{2k_B T}{m\omega_0 Q z_0^2}, \tag{12}$$

which yields an Allan variance $\sigma^2_{\mathrm{Allan}}(t_{\mathrm{int}}) = S_{\delta\omega\delta\omega}(\omega_0)/(2t_{\mathrm{int}}) = k_B T/(t_{\mathrm{int}} m\omega_0 Q z_0^2)$, a standard measure of frequency stability over the integration time $t_{\mathrm{int}}$ [47, 48] . To resolve the variance produced by the spins, we require $\sigma^2_{\delta\omega} > \sigma^2_{\mathrm{Allan}}(t_{\mathrm{int}})$. In the example of Fig. 2(b), single nuclear spin detection requires an integration time of $t_{\mathrm{int}} = 12\,\mathrm{min}$ to resolve the spin's variance.

## 5    Discussion

Our results show that statistical polarization can enable spin detection via dynamical backaction, providing far larger signals than the corresponding Boltzmann polarization for small spin ensembles $N < 10^6$. Nevertheless, for realistic samples two additional sources of spin decoherence need to be considered, resulting from spin-spin coupling and inhomogeneous broadening.

*Decoherence due to spin-spin coupling* – In typical nuclear magnetic resonance (NMR) experiments, interactions between neighboring nuclear spins can often be neglected when the Rabi frequency $\Omega_R$ exceeds the spin-spin coupling strength $J$. In that scenario, the range of Larmor frequencies that are affected by the spin lock is dominated by the spectral 'power broadening' equal to $\Omega_R$. By contrast, in the experiments we describe, the condition $\Omega_R \geq J$ is typically not fulfilled, and we are limited by $\Omega_R \ll 1/T_2, 1/T_1$, see condition (ii). As we cannot ignore spin-spin interaction in the weak-driving regime, $J$ is accounted for in the simulations through the spin decoherence time $T_2 = 100\,\mathrm{\mu s}$ [49].

*Inhomogeneous broadening* – Any realistic sample has a certain size and thus contains spins at various positions within the magnetic field gradient, resulting in a range of Larmor frequencies. For instance, a spin ensemble with a diameter $D = 100\,\mathrm{nm}$ in a gradient $G = 2\,\mathrm{MT\,m^{-1}}$ experiences fields over a range $\Delta B = D \times G = 0.2\,\mathrm{T}$. The ensemble's Larmor frequencies are spread over a spectral range $\gamma\Delta B \approx 8\,\mathrm{MHz}$ leading to inhomogeneous spectral broadening $1/T_2^* \simeq \gamma\Delta B$. If $T_2^*$ is shorter than the timescale of the spin-spin interaction, $T_2$ has to be replaced by $T_2^*$ in Eqs. (7) and (8), causing a broader and shallower signal distribution.

In our sample, the driving fields are necessarily weak to satisfy condition (ii), $\Omega_R = \gamma G_i z_0 \ll 1/T_2, 1/T_1$. As a consequence, only the spins within the narrow range $\omega_{\mathrm{L}} =$

$\omega_0 \pm \Omega_R$ are directly excited, yielding an inhomogeneous broadening of $1/T_2^* = \Omega_R$. By contrast, through spin-spin interactions, all the spins within $\omega_L = \omega_0 \pm 1/T_2$ are indirectly excited. As our method requires $\Omega_R \ll 1/T_2$, the broadening of the spin ensembles is limited by $1/T_2$, not $1/T_2^*$. This means that we need not be concerned about the effects of inhomogeneous broadening, as the spatial regions we excite are very small. Unfortunately, this narrow excited region also comes at a cost: it fundamentally limits the number of spins that can contribute to the signal. For example, in our case, the width of the slice in the $z$-gradient direction is approximately $\delta z \approx 0.25\,\mathrm{nm}$. While this small voxel size limits the available signal strength, it naturally leads to a high spatial selectivity, and thereby to excellent spatial resolution.

Indeed, the most exciting aspect of our method is the limit of probing a single nuclear spin, as demonstrated in Fig. 2. While the detection of a single *electron* spin with a silicon cantilever required an averaging time of roughly $4.7 \times 10^4\,\mathrm{s}$ in 2004 [50], our method offers the sensitivity for detecting a single *nuclear* spin (with a roughly $10^3$ times lower magnetic moment) in $12\,\mathrm{min}$. This value is found assuming that the resonator's frequency noise is dominated by thermomechanical fluctuations. Technical frequency noise (e.g., from temperature drift or laser absorption), can further increase the frequency noise and complicate spin detection. However, recent breakthroughs have achieved a $1\,\mathrm{mHz}$ dissipation-limited bandwidth [45] and improved frequency drift calibration [51]. These advances indicate that precise, stable, and long-term frequency measurements at the thermomechanical limit are possible.

In summary, we have presented a method for detecting nuclear spins using dynamical backaction in megahertz resonators. By focusing on statistical polarization, the approach enables single-spin sensitivity with simple hardware and no need for spin control. Our detection method uses a single drive (e.g. via electrical or optomechanical coupling) acting directly on the resonator. Our approach reduces the experimental overhead significantly compared to typical MRFM experiments, which require a microstrip in close proximity of the resonator [52] to generate periodic spin flipping through radio-frequency pulses [3]. Near-resonant spin-mechanics coupling also opens the possibility of coherently manipulating nuclear spins through mechanical driving [53]. An intriguing possibility arises when swapping the roles of the resonator and the spin ensemble for spin cooling through backaction [54], akin to cavity cooling in the reversed dissipation regime in cavity optomechanics [55, 56]. Our simplified study paves the way for delving into the intricacies of local spin dissipation and decoherence [57] and dipole-dipole interactions [58] in particular experimental configurations. It also lays the groundwork for exploring further opportunities of parametric driving [24] and multimode resonators [59–61]. With these capabilities, nanoscale MRI will become a versatile platform for nuclear spin quantum sensing and control on the atomic scale.

## Acknowledgments

We thank Raffi Budakian, Oded Zilberberg, and Jan Košata for fruitful discussions. This work was supported by the Swiss National Science Foundation (CRSII5_177198/1) and an ETH Zurich Research Grant (ETH-51 19-2).

# A   Analytical approach

## A.1   Langevin spin-membrane equations of motion

We offer further details on the analytical solution for the spin-mechanical model introduced in the main text. The model features a driven mechanical resonator moving along $z$, influencing an ensemble of $N$ spins. The spins interact also with a spatially-dependent magnetic field. The combined dynamics is described by the Hamiltonian

$$\mathcal{H} = \frac{\hat{p}^2}{2m} + \frac{1}{2}m\omega_0^2\hat{q}^2 - F_0\hat{q}\cos(\omega_\mathrm{d}t) - \hbar\omega_\mathrm{L}\hat{I}_z - \hbar\gamma\hat{q}\left(G_x\hat{I}_x + G_y\hat{I}_y + G_z\hat{I}_z\right), \tag{13}$$

where $\hbar$ is the reduced Planck constant, $\omega_0$ ($\omega_\mathrm{L}$) is the mechanical (Larmor) resonance frequency, $\omega_\mathrm{d}$ is the driving frequency, $G_i$ is the magnetic gradient along $i \in [x, y, z]$, $\gamma$ is the gyromagnetic ratio of a nuclear spin, and $F_0$ is the driving force. Here $\hat{q}$ and $\hat{p}$ stand for the position and momentum operators for the resonator. The spins are described by the collective spin operators $\hat{I}_i = \sum_{k=1}^{N} \hat{\sigma}_{i,k}/2$, where $\hat{\sigma}_{i,k}$ are the Pauli matrices describing a spin-$\frac{1}{2}$.

We extract the dissipative equations of motion (EOM) using the Heisenberg picture. We account for mechanical damping ($\Gamma_\mathrm{m}$) as well as spin decay ($T_1$) and decoherence ($T_2$). Furthermore, we consider thermomechanical noise, acting on the resonator, and polarization noise, parametrized by operators $\hat{\xi}(t)$ and $\hat{\zeta}_0(t)$, respectively. The corresponding Heisenberg-Langevin equations [62] read:

$$\ddot{\hat{q}} = -\omega_0^2\hat{q} - \Gamma_\mathrm{m}\dot{\hat{q}} + \frac{F_0}{m}\cos(\omega_\mathrm{d}t) + \frac{\hbar\gamma}{m}\left(G_x\hat{I}_x + G_y\hat{I}_y + G_z\hat{I}_z\right) + \hat{\xi}(t), \tag{14}$$

$$\dot{\hat{I}}_{x,y} = -\frac{1}{T_2}\hat{I}_{x,y} \pm \omega_\mathrm{L}\hat{I}_{y,x} \pm \gamma G_z\hat{q}\hat{I}_{y,x} \mp \gamma G_{y,x}\hat{q}\hat{I}_z, \tag{15}$$

$$\dot{\hat{I}}_z = \frac{1}{T_1}\left(\hat{\zeta}_0(t) - \hat{I}_z\right) - \gamma G_x\hat{q}\hat{I}_y + \gamma G_y\hat{q}\hat{I}_x, \tag{16}$$

where we identify the renormalized mechanical frequency as $\omega_0 \mapsto \sqrt{\omega_0^2 + \Gamma_\mathrm{m}^2/4}$.

Polarization noise is split into average and fluctuating contributions: $\hat{\zeta}_0(t) = I_0 + \delta\hat{I}_0(t)$. Here $I_0$ stands for the Boltzmann (thermal) equilibrium polarization [32]

$$I_0 = -N\left[(2I+1)\coth\left((2I+1)\hbar\omega_\mathrm{L}/(2k_BT)\right) - \coth(\hbar\omega_\mathrm{L}/(2k_BT))\right]/2, \tag{17}$$

with $N$ the number of spins in the considered ensemble and $I = \frac{1}{2}$ the spin number.

The resonator motion is driven well above its zero-point fluctuation. We can therefore apply a semiclassical approximation, which reduces the operators to real amplitudes, $\hat{q} \mapsto q$ and $\hat{I}_i \mapsto I_i$, yielding the equations of motion in the main text, Eqs. (4)-(6):

$$\ddot{q} = -\omega_\mathrm{d}^2q - \Gamma_\mathrm{m}\dot{q} + \frac{F_0}{m}\cos(\omega_\mathrm{d}t) + \frac{\hbar\gamma}{m}\mathbf{G}\cdot\mathbf{I} + \xi(t), \tag{18}$$

$$\dot{I}_{x,y} = -\frac{1}{T_2}I_{x,y} \pm (\omega_\mathrm{L} + \gamma qG_z)I_{y,x} \mp \gamma qG_{y,x}I_z, \tag{19}$$

$$\dot{I}_z = \frac{1}{T_1}\left(\zeta_0(t) - I_z\right) - \gamma q\left(G_xI_y - G_yI_x\right), \tag{20}$$

with $\mathbf{G} = (G_x, G_y, G_z)$, $\mathbf{I} = (I_x, I_y, I_z)$, thermomechanical force $\xi(t)$ acting on the resonator, and fluctuating classical polarization $\zeta_0(t) = I_0 + \delta I_0(t)$. The classical fluctuations amplitudes have Gaussian statistics with correlators

$$\langle\xi(t)\xi(t')\rangle = \frac{2\Gamma_\mathrm{m}k_BT}{m}\delta(t - t'), \tag{21}$$

$$\langle\delta\zeta_0(t)\delta\zeta_0(t')\rangle = \sigma_{\delta I_0}^2 e^{-|t-t'|/\tau}, \tag{22}$$

where $T$ is the temperature of the mechanical bath, $\tau$ stands for the spin bath autocorrelation time, and variance $\sigma^2_{\delta I_0} = \frac{N}{4}$ [33, 34].

We first examine the deterministic dynamics, governed by

$$\ddot{\bar{q}} = -\omega_0^2 \bar{q} - \Gamma_{\mathrm{m}} \dot{\bar{q}} + \frac{F_0}{m} \cos(\omega_{\mathrm{d}} t) + \frac{\hbar \gamma}{m} \mathbf{G} \cdot \bar{\mathbf{I}}, \tag{23}$$

$$\dot{\bar{I}}_{x,y} = -\frac{1}{T_2} \bar{I}_{x,y} \pm (\omega_{\mathrm{L}} + \gamma \bar{q} G_z) \bar{I}_{y,x} \mp \gamma \bar{q} G_{y,x} \bar{I}_z, \tag{24}$$

$$\dot{\bar{I}}_z = \frac{1}{T_1} \left( I_0 - \bar{I}_z \right) - \gamma \bar{q} \left( G_x \bar{I}_y - G_y \bar{I}_x \right), \tag{25}$$

where $\cdot\bar{\phantom{x}}\cdot$ denotes averages. These equations can also be found from averaging the Heisenberg EOM Eq. (14), Eq. (15) and Eq. (16), under the mean-field approximation, where cross-correlations are neglected, i.e., $\langle \hat{q} \hat{I}_i \rangle = \bar{q} \bar{I}_i$. We then examine how fluctuations induced by $\zeta_0(t)$ affect the system's dynamics.

## A.2   Slow-flow equations of motion

We further analyze here the deterministic solution to the main text Eqs. (4)-(6). We assume that the resonator dynamic is dominated by the external force. Thus, the coupling to the spins act as a small correction. We can then write the mechanical motion as $q(t) = q^{(0)}(t) + \delta q(t)$, where $q^{(0)}(t) = u_q^{(0)} \cos(\omega_{\mathrm{d}} t) + v_q^{(0)} \sin(\omega_{\mathrm{d}} t)$ with

$$u_q^{(0)} = \frac{F_0}{m \left[ \left( \omega_0^2 - \omega_{\mathrm{d}}^2 \right)^2 + \omega_{\mathrm{d}}^2 \Gamma_{\mathrm{m}}^2 \right]} \left( \omega_0^2 - \omega_{\mathrm{d}}^2 \right), \tag{26}$$

$$v_q^{(0)} = \frac{F_0}{m \left[ \left( \omega_0^2 - \omega_{\mathrm{d}}^2 \right)^2 + \omega_{\mathrm{d}}^2 \Gamma_{\mathrm{m}}^2 \right]} \omega_{\mathrm{d}} \Gamma_{\mathrm{m}}. \tag{27}$$

The part accounting for the spins then obeys

$$\ddot{\delta q} = -\omega_0^2 \delta q - \Gamma_m \dot{\delta q} + \frac{\hbar \gamma}{m} \mathbf{G} \cdot \mathbf{I}. \tag{28}$$

We express the solution for $\delta q(t)$ in terms of an ansatz

$$\delta q(t) = \delta a_q(t) + \delta u_q(t) \cos(\omega_{\mathrm{d}} t) + \delta v_q(t) \sin(\omega_{\mathrm{d}} t), \tag{29}$$

where $\delta a_q(t)$, $\delta u_q(t)$ and $\delta v_q(t)$ are real time-dependent amplitudes to be found. Employing this form of the solution is particularly beneficial when examining perturbations associated with the behavior of a driven harmonic oscillator. The dynamics of the spins in response to the mechanical motion can be calculated employing a similar ansatz

$$I_i(t) = a_i(t) + u_i(t) \cos(\omega_{\mathrm{d}} t) + v_i(t) \sin(\omega_{\mathrm{d}} t), \tag{30}$$

with amplitudes $a_i(t), u_i(t), v_i(t)$. Given Eq. (30), the spins exert a time-dependent force on the resonator given by

$$\delta F(t) = \hbar \gamma \mathbf{G} \cdot \mathbf{I}(t) = \hbar \gamma \left[ \mathbf{G} \cdot \mathbf{a}(t) + \mathbf{G} \cdot \mathbf{u}(t) \cos(\omega_d t) + \mathbf{G} \cdot \mathbf{v}(t) \sin(\omega_d t) \right], \tag{31}$$

where we used vector notation for $a_{x,y,z}(t), u_{x,y,z}(t), v_{x,y,z}(t)$.

At this stage, we have not yet introduced any constraints or approximations in the ansatz amplitudes. However, the calculation of the corrections $\delta a_q, \delta u_q, \delta v_q$ is greatly facilitated by assuming the weak impact of the spin-dependent force on the resonator.

Namely, we assume $\langle\langle|\hbar\gamma\mathbf{G}\cdot\mathbf{I}|\rangle\rangle_{T_d} \ll F_0$, where $\langle\langle...\rangle\rangle_{T_d}$ denotes the average over a drive period $T_\mathrm{d} = 2\pi/\omega_\mathrm{d}$. In this setting, we can assume the amplitudes $\delta a_q(t), \delta u_q(t), \delta v_q(t)$ with respect to $T_\mathrm{d}$, accounting for the transient evolution of the amplitude and phase of the resonator towards the steady state [63].

In the steady state, resonator and spin precession amplitudes settle to constant values, i.e. $\delta\dot{a}_q = \delta\dot{u}_q = \delta\dot{v}_q = 0$ and $\dot{a}_i = \dot{u}_i = \dot{v}_i = 0$. In our ansatz $q(t)$ thus acts as a harmonic magnetic field with frequency $\omega_\mathrm{d}$, acting on the spins. In particular, the spin prompts a spin precession component at frequency $\omega_\mathrm{d}$, according to Eq. (30). Note the ansatz does not presuppose the synchronization or "locking" of the spin dynamics with the external field. We seek if such a steady state can exist. To this end, we insert the ansatz for $q(t)$ and $I_i(t)$ in the mean-field equations of motion and equate the harmonic amplitudes at both sides of the equations with the same time dependence, a procedure dubbed the "harmonic balance" [36]. This approach also neglects super-harmonic generation (e.g. terms $\cos(2\omega_\mathrm{d}t)$, $\sin(2\omega_\mathrm{d}t)$) that arises from the mechanical motion driving the spins, which requires extending the harmonic ansatz for $q(t), I_i(t)$ to higher frequencies. Note that harmonic balance relies on the slowly-flowing nature of the amplitudes $a_i, u_i, v_i$ [63].

### A.2.1 Linear response theory: Deterministic dynamics

The introduction of the ansatz results in nonlinear couplings between the harmonic amplitudes of the mechanical resonator and the spins. The system's steady states are defined by the roots of these coupled polynomials. While we could solve these equations numerically using advanced algebraic methods, as detailed in reference [64] and implemented in the package [37], we opt for deriving an analytical solution within a linearized framework. Here we find the mechanical dynamics of the resonator in the weakly fluctuating regime $\langle\langle|\delta q|\rangle\rangle_{T_d} \ll \langle\langle|q^{(0)}|\rangle\rangle_{T_d}$. The smallness of $\delta q$ allows us to neglect the nonlinear coupling between the fluctuations $\delta u_q(t), \delta v_q(t)$ and the spin amplitudes $a_i(t), u_i(t), v_i(t)$. Under this linearization, the spin dynamics directly follows from the solutions of the first-order differential equations that do not contain $\delta a_q(t), \delta u_q(t), \delta v_q(t)$, namely

$$\dot{a}_{x,y} + \frac{1}{T_2}a_{x,y} \mp \omega_\mathrm{L}a_{y,x} = 0, \tag{32a}$$

$$\dot{u}_{x,y} + \frac{1}{T_2}u_{x,y} \mp \omega_\mathrm{L}u_{y,x} + \omega_\mathrm{d}v_{x,y} - \gamma u_q^{(0)}(G_{z,x}a_{y,z} - G_{y,z}a_{z,x}) = 0, \tag{32b}$$

$$\dot{v}_{x,y} + \frac{1}{T_2}v_{x,y} \mp \omega_\mathrm{L}v_{y,x} - \omega_\mathrm{d}u_{x,y} - \gamma v_q^{(0)}(G_{z,x}a_{y,z} - G_{y,z}a_{z,x}) = 0, \tag{32c}$$

$$\dot{a}_z + \frac{1}{T_1}a_z - I_0\frac{1}{T_1} = 0, \tag{32d}$$

$$\dot{u}_z + \frac{1}{T_1}u_z + \omega_\mathrm{d}v_z - \gamma u_q^{(0)}(G_ya_x - G_xa_y) = 0, \tag{32e}$$

$$\dot{v}_z + \frac{1}{T_1}v_z - \omega_\mathrm{d}u_z - \gamma v_q^{(0)}(G_ya_x - G_xa_y) = 0. \tag{32f}$$

The resonator features a high quality factor ($\Gamma_\mathrm{m} \ll 1/T_2, 1/T_1$) which, together with the weak spin-resonator coupling ($\gamma G_i z_0 \ll 1/T_2, 1/T_1$) lead to spins quickly reaching steady state compared to the slower resonator timescale. This condition permits the application of approximation methods, like adiabatic elimination of the spins [65], in order to approximate the time evolution of the resonator towards its steady state. Our focus is nevertheless on the global steady state behavior, where all amplitudes in the problem are fixed. Solving Eqs. (32) when $\dot{a}_i = \dot{u}_i = \dot{v}_i = 0$ to find the steady state amplitudes $\mathbf{a}|_{t\to\infty}, \mathbf{u}|_{t\to\infty}, \mathbf{v}|_{t\to\infty}$ leads to a steady state force

$$\delta F|_{t\to\infty} \approx \hbar\gamma\left[\mathbf{G}\cdot\mathbf{a}|_{t\to\infty} + \mathbf{G}\cdot\mathbf{u}_{t\to\infty}\cos(\omega_dt) + \mathbf{G}\cdot\mathbf{v}_{t\to\infty}\sin(\omega_dt)\right]. \tag{33}$$

Such force will not be in phase with the external resonator's driving (its quadratures will not be aligned with the drive), namely $u_q^{(0)}$, $v_q^{(0)}$. To facilitate the expressions, we choose a phase/time origin for the driven resonator (i.e. we perform a gauge fixing), such that $v_q^{(0)} = 0$ and $u_q^{(0)} = F_0/(m\sqrt{(\omega_0^2 - \omega_d^2)^2 + \omega_d^2\Gamma_m^2})$. In this gauge, we can identify $\delta F|_{t\to\infty} \approx \delta F_0 - \delta\Gamma_m \dot{q} - \delta\Omega^2 q$, where $\delta F_0 = \hbar\gamma I_0 G_z/m$ and

$$\delta\Gamma_m = \frac{\hbar\gamma^2 I_0 \left(G_x^2 + G_y^2\right)}{m} \frac{\omega_L T_2^{-1}}{\left(T_2^{-2} + \omega_d^2\right)^2 + 2\left(T_2^{-1} - \omega_d\right)\left(T_2^{-1} + \omega_d\right)\omega_L^2 + \omega_L^4}, \tag{34}$$

$$\delta\Omega^2 = -\frac{\hbar\gamma^2 I_0 \left(G_x^2 + G_y^2\right)}{m} \frac{\omega_L \left(T_2^{-2} - \omega_d^2 + \omega_L^2\right)}{\left(T_2^{-2} + \omega_d^2\right)^2 + 2\left(T_2^{-1} - \omega_d\right)\left(T_2^{-1} + \omega_d\right)\omega_L^2 + \omega_L^4}. \tag{35}$$

We can now reconstruct the mechanical evolution in the steady state from the effective equation of motion. Under resonant driving $\omega_d = \omega_0$,

$$\ddot{q} + (\Gamma_m + \delta\Gamma_m)\,\dot{q} + (\omega_0 + \delta\omega)^2\,q|_{t\to\infty} = F_0 \cos(\omega_d t) + \delta F_0, \tag{36}$$

with $\delta\omega = \frac{1}{2}\frac{\delta\Omega^2}{\omega_d}$. We can rewrite $\delta\Gamma_m$ and $\delta\omega$ in a more convenient way leading to Eqs. (7) and (8)

$$\bar{\delta\omega} = -g^2 \left(\frac{\omega_L + \omega_d}{T_2^{-2} + (\omega_L + \omega_d)^2} + \frac{\omega_L - \omega_d}{T_2^{-2} + (\omega_L - \omega_d)^2}\right) I_0, \tag{37}$$

$$\delta\bar{\Gamma}_m = -g^2 \left(\frac{T_2^{-1}}{T_2^{-2} + (\omega_L + \omega_d)^2} - \frac{T_2^{-1}}{T_2^{-2} + (\omega_L - \omega_d)^2}\right) I_0, \tag{38}$$

where $g^2 = \hbar\gamma^2 \left(G_x^2 + G_y^2\right)/(4m\omega_d)$.

## A.2.2 Linear response theory: Fluctuation dynamics

As the system relaxes, weak fluctuations have their strongest impact near the steady state. We therefore adopt a perturbative approach where the system remains close to equilibrium. This allows us to set the spin amplitudes at $t \to -\infty$ in Eq. (33) as linear functions of the fluctuating field $\delta I_0(t)$. This linearization around equilibrium ensures that noise effects remain analytically tractable: it makes Eq. (37) explicitly dependent on the fluctuating prefactor

$$g^2 \delta I_0(t) = \frac{\hbar\gamma^2 \left(G_x^2 + G_y^2\right)}{4m\omega_d}\delta I_0(t). \tag{39}$$

The standard deviation of the frequency shift, $\sigma_{\delta\omega}$, becomes proportional to that of $\delta I_0(t)$, namely,

$$\sigma_{\delta\omega} = \frac{\hbar\gamma^2 \sigma_{\delta I_0}\left(G_x^2 + G_y^2\right)}{4m\omega_d} \left(\frac{\omega_L + \omega_d}{T_2^{-2} + (\omega_L + \omega_d)^2} + \frac{\omega_L - \omega_d}{T_2^{-2} + (\omega_L - \omega_d)^2}\right). \tag{40}$$

From Eq. (9), the relation $|\sigma_{\delta\omega}/\bar{\delta\omega}| = |\sigma_{\delta I_0}/I_0|$ follows, consistent with standard uncertainty propagation under Gaussian or symmetric noise. This approach assumes the noise is regular and uncorrelated, even if $\sigma_{\delta I_0}$ is comparable to or larger than $I_0$ (within the validity of the linearization in Eqs. (32)), and that its evolution is slow, with correlation time $\tau$ much longer than the mechanical response time $2\pi/\Gamma$, allowing the resonator to track the spin force quasi-adiabatically. The key condition is a clear separation of timescales: when spin fluctuations evolve slowly compared to the mechanical response ($\tau \gg 2\pi/\Gamma$),

the resonator can adiabatically track the varying spin force. This justifies treating it as in quasi-steady state at each instant and motivates the phenomenological model used in the main text to capture how slow, yet sizable, fluctuations set $\sigma_{\delta\omega}$. Deviations from this limit are briefly noted below and discussed in more detail in the main text.

### A.2.3   Time-dependent polarization: beyond the adiabatic limit

The adiabatic approximation, in which time derivatives are set to zero in Eqs. (32), is exact when $I_0$ is constant, as in Boltzmann polarization. In what follows, we make this statement explicit by solving the full frequency-dependent problem and showing that, when $I_0(t) = I_0$, only the zero-frequency component of the spin response contributes, exactly recovering the steady-state result.

We now consider the more general case where $I_0$ varies in time. Rather than dropping time derivatives, we take the Fourier transform of Eqs. (32) with $I_0 \rightarrow I_0(t)$, using frequency $\omega$ conjugate to the slow time. This turns the equations into linear algebraic relations. The resulting spin amplitudes $\tilde{a}_{x,y,z}(\omega)$, $\tilde{u}_{x,y,z}(\omega)$, and $\tilde{v}_{x,y,z}(\omega)$ read

$$\tilde{a}_z(\omega) = \frac{i\,\tilde{I}_0(\omega)}{i + \omega/T_1},$$

$$\tilde{u}_x(\omega) = -\frac{i\,\tilde{I}_0(\omega)\,\gamma}{D(\omega)}\bigg[G_x\,\omega_L\left(-2i\,\omega\,\omega_d\,v_q^{(0)} + u_q^{(0)}(\omega^2 + \omega_d^2 - \omega_L^2)\right),$$

$$+ G_y\,\omega_L^2\,(-i\,\omega\,u_q^{(0)} + \omega_d\,v_q^{(0)}) + G_y\,(\omega^2 - \omega_d^2)\,(i\,\omega\,u_q^{(0)} + \omega_d\,v_q^{(0)})$$

$$+ \frac{1}{T_2}\Big(2G_x\,\omega_L\,(i\,\omega\,u_q^{(0)} + \omega_d\,v_q^{(0)}) + 2i\,G_y\,\omega\,\omega_d\,v_q^{(0)} + G_y\,u_q^{(0)}(-3\omega^2 + \omega_d^2 + \omega_L^2)$$

$$+ \frac{1}{T_2}\Big(-3i\,G_y\,\omega\,u_q^{(0)} - G_y\,\omega_d\,v_q^{(0)} - G_x\,\omega_L\,u_q^{(0)} + \frac{G_y\,u_q^{(0)}}{T_2}\Big)\Big)\bigg], \tag{41}$$

$$\tilde{u}_y(\omega) = \frac{\tilde{I}_0(\omega)\,\gamma}{D(\omega)}\bigg[G_x\,\omega_L^2\,(\omega\,u_q^{(0)} + i\,\omega_d\,v_q^{(0)})$$

$$- G_x\,(\omega^2 - \omega_d^2)\,(\omega\,u_q^{(0)} - i\,\omega_d\,v_q^{(0)}) + G_y\,\omega_L\,(-2\omega\,\omega_d\,v_q^{(0)} - i\,u_q^{(0)}(\omega^2 + \omega_d^2 - \omega_L^2))$$

$$+ \frac{1}{T_2}\Big(-2G_x\,\omega\,\omega_d\,v_q^{(0)} - i\,G_x\,u_q^{(0)}(3\omega^2 - \omega_d^2 - \omega_L^2)$$

$$+ 2G_y\,\omega_L\,(\omega\,u_q^{(0)} - i\,\omega_d\,v_q^{(0)}) + \frac{1}{T_2}\Big(3G_x\,\omega\,u_q^{(0)} - i\,G_x\,\omega_d\,v_q^{(0)} + i\,G_y\,\omega_L\,u_q^{(0)} + \frac{i\,G_x\,u_q^{(0)}}{T_2}\Big)\Big)\bigg], \tag{42}$$

$$\tilde{v}_x(\omega) = \frac{\tilde{I}_0(\omega)\,\gamma}{D(\omega)}\bigg[G_x\,\omega_L\left(2\omega\,\omega_d\,u_q^{(0)} - i\,v_q^{(0)}(\omega^2 + \omega_d^2 - \omega_L^2)\right)$$

$$+ G_y\,\omega_L^2\,(-\omega\,v_q^{(0)} + i\,\omega_d\,u_q^{(0)}) + G_y\,(\omega^2 - \omega_d^2)\,(\omega\,v_q^{(0)} + i\,\omega_d\,u_q^{(0)})$$

$$+ \frac{i}{T_2}\Big(2G_x\,\omega_L\,(-i\,\omega\,v_q^{(0)} + \omega_d\,u_q^{(0)}) + 2i\,G_y\,\omega\,\omega_d\,u_q^{(0)} + G_y\,v_q^{(0)}(3\omega^2 - \omega_d^2 - \omega_L^2)$$

$$+ \frac{1}{T_2}\Big(G_x\,\omega_L\,v_q^{(0)} + 3i\,G_y\,\omega\,v_q^{(0)} - G_y\,\omega_d\,u_q^{(0)} - \frac{G_y\,v_q^{(0)}}{T_2}\Big)\Big)\bigg], \tag{43}$$

$$\tilde{v}_y(\omega) = -\frac{i\,\tilde{I}_0(\omega)\,\gamma}{D(\omega)}\bigg[G_x\,\omega_L^2\,(i\,\omega\,v_q^{(0)} + \omega_d\,u_q^{(0)})$$

$$+ G_x\,(\omega^2 - \omega_d^2)\,(-i\,\omega\,v_q^{(0)} + \omega_d\,u_q^{(0)}) + G_y\,\omega_L\,(2i\,\omega\,\omega_d\,u_q^{(0)} + v_q^{(0)}(\omega^2 + \omega_d^2 - \omega_L^2))$$

$$- \frac{1}{T_2}\Big(-2i\,G_x\,\omega\,\omega_d\,u_q^{(0)} + G_x\,v_q^{(0)}(-3\omega^2 + \omega_d^2 + \omega_L^2),$$

$$+ 2G_y\,\omega_L\,(-i\,\omega\,v_q^{(0)} + \omega_d\,u_q^{(0)}) + \frac{1}{T_2}\Big(-3i\,G_x\,\omega\,v_q^{(0)} + G_x\,\omega_d\,u_q^{(0)} + G_y\,\omega_L\,v_q^{(0)} + \frac{G_x\,v_q^{(0)}}{T_2}\Big)\Big)\bigg], \tag{44}$$

where $u_q^{(0)}$ and $v_q^{(0)}$ are the driven resonator quadratures from Eqs. (26) and $D(\omega)$ stands for a denominator

$$D(\omega) = \left(\omega + \frac{i}{T_1}\right)\left(\omega - \omega_d - \omega_L + \frac{i}{T_2}\right)\left(\omega + \omega_d - \omega_L + \frac{i}{T_2}\right) \times$$

$$\left(\omega - \omega_d + \omega_L + \frac{i}{T_2}\right)\left(\omega + \omega_d + \omega_L + \frac{i}{T_2}\right). \tag{45}$$

Moreover, $\tilde{a}_x(\omega) = \tilde{a}_y(\omega) = 0 = \tilde{u}_z(\omega) = \tilde{v}_z(\omega) = 0$. Equations (41) show that the nonzero amplitudes are given by the product of the spin susceptibility, peaked at $\omega_d \pm \omega_L$, with linewidths given by $T_1^{-1}$ and $T_2^{-1}$, and $\tilde{I}(\omega)$.

The spin force, exerted on the resonator, can be expressed, as a time-dependent version of Eq. (31), by inverse-transforming the frequency-domain expressions into the envelopes $a_{x,y,z}(t)$, $u_{x,y,z}(t)$, and $v_{x,y,z}(t)$. For constant (Boltzmann) polarization $I_0$, the spectrum reduces to $\tilde{I}_0(\omega) = 2\pi\delta(\omega)$, so only the spin susceptibility at $\omega = 0$ contributes. This recovers the steady-state result of Eq. (33), independent of $T_1$. To build intuition for a time-varying $I_0(t)$, consider a slowly varying deterministic signal expanded as a Fourier series $I_0(t) = \frac{1}{2}\sum_k I_k e^{-i\omega_k t} + I_k^* e^{i\omega_k t}$, which frequency representation

$$\tilde{I}_0(\omega) = \frac{1}{4\pi}\sum_k I_k\delta(\omega - \omega_k) + I_k^*\delta(\omega + \omega_k), \tag{46}$$

where $\omega_k \ll \omega_d$ to ensure consistency with the slow-flow ansatz (30). This implies, for example, $a_{x,y,z}(t) = (1/2\pi)\sum_k e^{-i\omega_k t}\tilde{a}_{x,y,z}(\omega_k) + \text{c.c}$ and similarly for $u_{x,y,z}(t)$, and $v_{x,y,z}(t)$. The spin force then reads

$$\delta F(t) \approx \frac{\hbar\gamma}{4\pi}\sum_k \mathbf{G} \cdot \left[\tilde{\mathbf{a}}(\omega_k) + \tilde{\mathbf{u}}(\omega_k)\cos(\omega_d t) + \tilde{\mathbf{v}}(\omega_k)\sin(\omega_d t)\right]e^{-i\omega_k t} + \text{c.c.}, \tag{47}$$

where $\tilde{\mathbf{a}}(\omega)$, $\tilde{\mathbf{u}}(\omega)$, $\tilde{\mathbf{v}}(\omega)$ group the $x, y, z$ components, and c.c. denotes the complex conjugate. These amplitudes satisfy $\tilde{\mathbf{a}}^*(-\omega) = \tilde{\mathbf{a}}(\omega)$, and similarly for $\tilde{\mathbf{u}}$ and $\tilde{\mathbf{v}}$. Note that, for $\omega_k \neq 0$, these amplitudes depend explicitly on $T_1$.

Equation (47) shows that $\delta F(t)$ inherits the frequency content of $\tilde{I}_0(\omega)$, filtered by the spin susceptibility at each frequency $\omega = \omega_k$ [cf. Eqs. (41)]. The result is a slow envelope at $\omega_k \ll \omega_d$ modulating the carrier at $\omega_d$.

The expression (47) can be succintly written in integral form by defining the spectral densities $G^a(\omega) \equiv \sum_k \mathbf{G} \cdot \tilde{\mathbf{a}}(\omega_k)\,\delta(\omega - \omega_k)$, $G_\pm^w(\omega) = \sum_k \mathbf{G} \cdot (\tilde{\mathbf{u}}(\omega_k) \mp i\,\tilde{\mathbf{v}}(\omega_k))\delta(\omega - \omega_k),$.

$$\delta F(t) \approx \frac{\hbar\gamma}{4\pi}\int_{-\infty}^{\infty} d\omega \left[G^a(\omega)e^{-i\omega t} + G_+^w(\omega)e^{-i(\omega+\omega_d)t} + G_-^w(\omega)e^{-i(\omega-\omega_d)t}\right] + \text{c.c.} \tag{48}$$

Since $G^a(\omega), G_\pm^w(\omega) \propto \tilde{I}_0(\omega)$, Eq. (48) the spin-force linewidth is set by the linewidth of $\tilde{I}_0(\omega)$, with susceptibility filtering out components $|\omega| \gtrsim 1/T_1$. If $\Gamma \gg \omega_k$, the resonator follows the force quasi-adiabatically, so its frequency shift and damping sample the full susceptibility. Back to the Fourier domain,

$$\delta F(\Omega) \approx \frac{\hbar\gamma}{2}G^a(0)e^{-i\omega t} + G_+^w(\Omega + \omega_d) + G_-^w(\Omega - \omega_d)\right] + \text{c.c.}. \tag{49}$$

Promoting $I_0(t)$ to a stationary stochastic process, makes $G^a(\omega)$ and $G_\pm^w(\omega)$ random variables that are linear in the Fourier amplitudes $\tilde{I}_0(\omega)$:

$$G^a(\omega) = H_a(\omega), \tilde{I}_0(\omega), \quad G_+^w(\omega) = H_+(\omega), \tilde{I}_0(\omega), \quad G_-^w(\omega) = H_-(\omega), \tilde{I}_0(\omega), \tag{50}$$

where $H_{a,\pm}(\omega)$ are the spin-susceptibility functions in Eqs. (41). The input polarization power spectral density (PSD), arising from a local spin bath, acting on the relevant spins coupled to the resonator, is defined by

$$\langle \tilde{I}_0(\omega), \tilde{I}_0^*(\omega') \rangle = 2\pi\delta(\omega - \omega')S_{I_0}(\omega), \tag{51}$$

where

$$S_{I_0,I_0}(\omega) \propto \frac{\tau}{1 + \omega^2\tau^2}. \tag{52}$$

for an Ornstein–Uhlenbeck process with correlation time $\tau$.

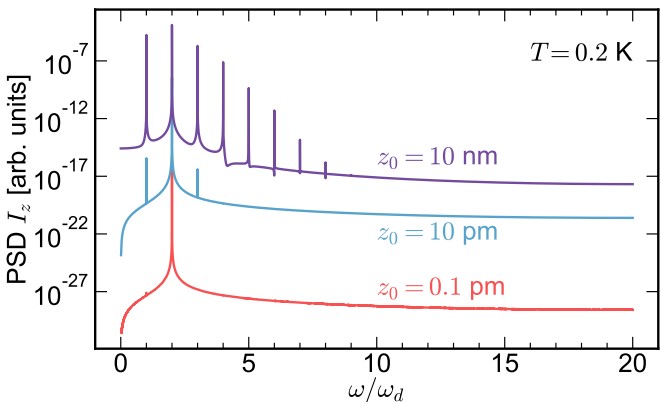

Figure 4: Power spectral density (PSD) of the steady state of a simulated SiN membrane resonator with zero frequency component removed [10] for different driving amplitudes $z_0$, $m = 5 \times 10^{-12}$ kg, $\omega_0/2\pi = 1.4$ MHz, $G_x = G_y = 2$ MT/m, $G_z = 1$ MT/m, $N = 10^6$ spins, $T_2 = 100\,\mu$s, $T_1 = 50$ ms and resonant driving of the resonator $\omega_d = \omega_0$.

From Eqs.(49)-(52) and assuming the sideband channels are uncorrelated, the force PSD reads

$$S_{\delta F, \delta F}(\Omega) \approx \left(\frac{\hbar \gamma}{2}\right)^2 \left[ |H_a(\Omega)|^2 S_{I_0, I_0}(\Omega) + \sum_{p=\pm} |H_p(\Omega + p\omega_d)|^2 S_{I_0, I_0}(\Omega + p\omega_d) \right]. \quad (53)$$

Equation (53) shows that the force-noise bandwidth is set by $1/\tau$, while the spin susceptibility shapes the weighting across frequencies and generates sidebands at $\pm\omega_d$. In the quasi-adiabatic limit ($\Gamma$ much larger than the polarization bandwidth), the resonator tracks these fluctuations, so both its frequency shift and damping sample the full spin susceptibility across frequency. This motivates the simplified model employed in main text Sec. 4.

### A.2.4   Beyond linear response

For certain parameter regimes, the nonlinearities in Eqs. (18)-(20) can lead to complex behavior in the stationary limit $t \to \infty$, including self-sustained motion, multi-stability, and limit cycles [66, 67]. In particular, the analogy with optomechanics is expected to break down when the Rabi frequency is comparable to the spin dissipation: $\gamma G_i z_0 \sim 1/T_2, 1/T_1$. In that case, the spins' equilibration is not fast enough before they act back on the resonator, and spin-resonator timescales cannot be adiabatically separated. Effectively, the resonator motion then triggers spin-induced nonlinear effects, such as a periodic time modulation of the Larmor frequency due to the $G_z$ gradient (see main text Eqs. (4)-(6)), with frequency $\omega_d$. The resonator's response would then pick up higher frequency components not described by Eq. (30). While under the linearized theory, the steady state value is time independent and equal to $I_z = I_0$, we observe the generation of higher order harmonics in the spectrum of $I_z$ [Fig. 4]. Note that in our simulations, we do not focus on the regime where higher excitation makes the spin-conservation constraint ($d(\sum_i I_i^2)/dt = 0$) relevant, which would lead to additional 'many-wave mixing'.

A comprehensive examination of nonlinearities in our detection protocol is beyond the scope of this manuscript; however, we offer a brief overview of the necessary approach below. The impact of weak nonlinearities can be expressed by still expanding the solution

for $x(t)$ with an ansatz of the form

$$x(t) = a_q(t) + u_q(t)\cos(\omega_{\rm d}t) + v_q(t)\sin(\omega_{\rm d}t), \tag{54}$$

which includes both the displacement by the driving field and small fluctuations, while keeping the same ansatz for the spins in Eq. (30). We will account now for the nonlinear corrections to this resonant behavior. The equations of motion for the ansatz amplitudes without linearization read

$$\omega_0^2 a_q + \tilde{G}_x a_x + \tilde{G}_y a_y + \tilde{G}_z a_z + \Gamma_{\rm m}\dot{a}_q = 0, \tag{55}$$

$$\omega_0^2 u_q - \omega_{\rm d}^2 u_q + \Gamma_{\rm m}\omega_{\rm d} v_q + 2\omega_{\rm d}\dot{v}_q - F_0 + \tilde{G}_x u_x + \tilde{G}_y u_y + \tilde{G}_z u_z + \Gamma_{\rm m}\dot{u}_q = 0, \tag{56}$$

$$\omega_0^2 v_q - \omega_{\rm d}^2 v_q - 2\omega_{\rm d}\dot{u}_q - \Gamma_{\rm m}\omega_{\rm d} u_q \tilde{G}_x v_x + \tilde{G}_y v_y + \tilde{G}_z v_z + \Gamma_{\rm m}\dot{v}_q = 0, \tag{57}$$

for the membrane motion, and

$$\frac{1}{T_2}a_x + \omega_{\rm L}a_y + \tilde{G}_z a_q a_y - \tilde{G}_y a_q a_z - \frac{\tilde{G}_y}{2}(u_q u_z - v_q v_z) + \frac{\tilde{G}_z}{2}(u_q u_y + v_q v_y) + \dot{a}_x = 0, \tag{58}$$

$$\frac{1}{T_2}u_x + \omega_{\rm L}u_y + \omega_{\rm d}v_x + \tilde{G}_z a_q u_y + \tilde{G}_z a_y u_q - \tilde{G}_y a_q u_z - \tilde{G}_y a_z u_q + \dot{u}_x = 0, \tag{59}$$

$$\frac{1}{T_2}v_x + \omega_{\rm L}v_y - \omega_{\rm d}u_x + \tilde{G}_z a_q v_y + \tilde{G}_z a_y v_q - \tilde{G}_y a_z v_q - \tilde{G}_y a_q v_z + \dot{v}_x = 0, \tag{60}$$

$$\frac{1}{T_2}a_y - \omega_{\rm L}a_x + \tilde{G}_x a_q a_z - \tilde{G}_z a_q a_x + \frac{\tilde{G}_x}{2}(u_q u_z + v_q v_z) - \frac{\tilde{G}_z}{2}(v_q v_x - u_q u_x) + \dot{a}_y = 0, \tag{61}$$

$$\frac{1}{T_2}u_y + \omega_{\rm d}v_y - \omega_{\rm L}u_x + \tilde{G}_x a_q u_z + \tilde{G}_x a_z u_q - \tilde{G}_z a_q u_x - \tilde{G}_z a_x u_q + \dot{u}_y = 0, \tag{62}$$

$$\frac{1}{T_2}v_y - \omega_{\rm d}u_y + \tilde{G}_x a_z v_q - \omega_{\rm L}v_x + \tilde{G}_x a_q v_z - \tilde{G}_z a_x v_q - \tilde{G}_z a_q v_x + \dot{v}_y = 0, \tag{63}$$

$$\frac{1}{T_1}a_z - I_0\frac{1}{T_1} + \tilde{G}_y a_q a_x - \tilde{G}_x a_q a_y - \frac{\tilde{G}_x}{2}(u_q u_y - v_q v_y) + \frac{\tilde{G}_y}{2}(u_q u_x + v_q v_x) + \dot{a}_z = 0, \tag{64}$$

$$\frac{1}{T_1}u_z + \omega_{\rm d}v_z + \tilde{G}_y a_q u_x + \tilde{G}_y a_x u_q - \tilde{G}_x a_y u_q - \tilde{G}_x a_q u_y + \dot{u}_z = 0, \tag{65}$$

$$\frac{1}{T_1}v_z - \omega_{\rm d}u_z + \tilde{G}_y a_q v_x + \tilde{G}_y a_x v_q - \tilde{G}_x a_q v_y - \tilde{G}_x a_y v_q + \dot{v}_z = 0. \tag{66}$$

for the spin components. Note the shorthand $\tilde{G}_i = \gamma G_i$.

The resonator's susceptibility can then by found by (i) finding the steady states of these equations, i.e. finding the roots of a system of coupled polynomials arising from $\dot{a}_i = \dot{u}_i = \dot{v}_i = \dot{a}_q = \dot{u}_q = \dot{v}_q = 0$, and (ii) performing linear fluctuation analysis around these solutions. These two steps can be facilitated by the use of the HarmonicBalance.jl package [37].

The frequency spectrum in Fig. 4 reveals that as the driving strength increases, the lowest order nonlinear effect is the generation of a second harmonic at a frequency $2\omega_{\rm d}$. Similar equations to Eqs. (55)-(66) can be similarly obtained for the amplitudes of an extended ansatz that includes also the higher harmonic generated at $2\omega_{\rm d}$.

Considering fluctuation dynamics, significant non-Gaussian deviations, arising from nonlinear effects, become more relevant as noise strength increases, requiring higher-order corrections to accurately describe frequency shift statistics. For sufficiently large noise, activation between multiple stationary states may also occur, further modifying the system's response. The analysis of these effects falls outside of the scope of the current study.

## B   Relaxation

The spin lifetime $T_1$ of nuclear spins resulting from energy relaxation can vary strongly in typical nuclear magnetic resonance (NMR) experiments, ranging from microseconds to days. Our experimental situation is untypical, as we will probe nanoscale samples at low temperatures and low magnetic fields. We do not need a very specific value for $T_1$, as our analytical results hold as long as $\Gamma_m \ll 1/T_1$. To avoid speculation about the dependency of $T_1$ on field strength and temperatures below $70\,\mathrm{K}$, we use the same value of $T_1 = 50\,\mathrm{ms}$ for all our simulations. If needed for specific experimental situation, we envisage reducing $T_1$ by introducing paramagnetic agents, such as free radicals or metal ions [68].

## C   Exact numerical simulations

To verify our theoretical predictions, we wish to numerically simulate our mean-field EOM given by Eqs. (18)-(20). Due to the large span of magnitude of our problem (entailed in condition (i) the spins' force on the resonator, $\delta F$, is substantially weaker than the driving force, i.e., $|\delta F| \ll F_0$), we wish to rewrite the equations in a displaced frame where we simulate the fluctuations/deviations from the bare driven harmonic oscillator. We can use the ansatz $q(t) = q_0(t) + \delta q(t)$, where $q_0(t)$ is the steady-state solution of a bare driven harmonic oscillator (without spins), given by:

$$q_0 = \frac{F_0}{m\sqrt{(-\omega_d^2 + \omega_0^2)^2 + \omega_d^2\Gamma_m^2}}\cos\left(\omega_d t + \phi\right), \tag{67}$$

$$\phi = \arctan\left(-\frac{\omega_d\Gamma_m}{-\omega_d^2 + \omega_0^2}\right). \tag{68}$$

From here we can rewrite Eqs. (18)-(20) without the driving term:

$$\ddot{\delta q} = -\omega_0^2\delta q - \Gamma_m\dot{\delta q} - \frac{\hbar\gamma}{m}\mathbf{G}\cdot\mathbf{I} + \xi(t), \tag{69}$$

$$\dot{I}_x = -\frac{1}{T_2}I_x - \omega_L I_y + \gamma(q_0 + \delta q)\left(G_y I_z - G_z I_y\right), \tag{70}$$

$$\dot{I}_y = -\frac{1}{T_2}I_y + \omega_L I_x + \gamma(q_0 + \delta q)\left(G_z I_x - G_x I_z\right), \tag{71}$$

$$\dot{I}_z = \frac{1}{T_1}\left(\zeta_0(t) - I_z\right) + \gamma(q_0 + \delta q)\left(G_x I_y - G_y I_x\right). \tag{72}$$

Eqs. (69)-(72) describe the system in the laboratory frame. We can now solve them numerically using an explicit Runge-Kutta method of order 8, which is well-suited for handling the large separation of timescales in the problem [43]. In our simulations, we use a reduced (effective) quality factor $Q_{\mathrm{eff}}$ to reduce the simulation time. As the model does not explicitly depends on $\Gamma_m$, the influence of the resonator's quality factor is limited. Although, we need to keep in mind condition (iii) imposing $\Gamma_m \ll 1/T_1, T_2$. Interestingly, we notice that for $\Gamma_m \gtrsim 1/T_1$, the numerical simulations still lie very close to the analytical model. As an example, Fig. 2(a) is obtained with a simulated quality factor $Q_{\mathrm{eff}} = 2 \cdot 10^4$ giving $\Gamma_m = 2\pi \times 275\,\mathrm{Hz}$ whereas $1/T_1 = 20\,\mathrm{Hz}$. Note, however, that this does not apply to $\Gamma_m \ll 1/T_2$. We used Eqs. (69)-(72) for the Boltzmann polarization case.

However, this approach is not satisfactory for very long time scales as required for the statistical polarization case where we want to simulate for multiple correlation times (for example $t_{\mathrm{final}} = 100\tau$). Indeed, to properly resolve the Larmor precession, the timestep $\Delta t$

is chosen to be 40 times smaller than the precession period, i.e. $\Delta t = 2\pi/(40\omega_L) \sim 5\,\text{ns}$. For a simulation of length $t_{\text{final}} = 100\tau = 5\,\text{s}$, this requires 1 billion points to extract the mean and variance of one trajectory. This is obviously a massive limitation to explore the effect of parameters on the final frequency shift. To speed up our simulation, we use a version of Eqs.(18)-(20) where the spins are in a rotating frame at the Larmor frequency $\omega_L$ and where the mechanics is in a frame rotating at the driving frequency $\omega_d$. In these frames, the spin precession as well as the mechanical motion is quasi static. The fastest frequency is now given by $1/T_2 = 10\,\text{kHz}$, giving now a timestep $\Delta t = T_2/40 = 2.5\,\mu\text{s}$, reducing the amount of points by almost 3 orders of magnitude. The downside of going to a rotating frame is that we have to neglect the effect of the magnetic gradient in the $z$ direction $(G_z)$. However, we realized that the effect of the aforementioned gradient is to produce a frequency jittering of the Larmor frequency (due to the displacement of the resonator across it, modifying the Larmor frequency of the spins). This effect is, nevertheless, completely negligible compared to the frequency variance generated by the statistical polarization of the spins.

## C.1 Rotating frame

We first present the rotating frame transformation for the mechanical resonator, we rewrite $\delta q$ as

$$\delta q = \widetilde{\delta q_1} \cos(\omega_d t) + \widetilde{\delta q_2} \sin(\omega_d t), \tag{73}$$

with now $\widetilde{\delta q_1}$ and $\widetilde{\delta q_2}$ the quasi-static in-phase and quadrature components of the mechanical displacement. The resonator EOM can be written as:

$$\ddot{\widetilde{\delta q_1}} = -2\dot{\widetilde{\delta q_2}}\omega_d - \Gamma_m(\dot{\widetilde{\delta q_1}} + \widetilde{\delta q_2}\omega_d) + 2\frac{\gamma\hbar}{m}(\mathbf{G}\cdot\mathbf{I})\cos(\omega_d t) + \widetilde{\xi}(t), \tag{74}$$

$$\ddot{\widetilde{\delta q_2}} = 2\dot{\widetilde{\delta q_1}}\omega_d - \Gamma_m(\dot{\widetilde{\delta q_2}} - \widetilde{\delta q_1}\omega_d) + 2\frac{\gamma\hbar}{m}(\mathbf{G}\cdot\mathbf{I})\sin(\omega_d t) + \widetilde{\xi}(t), \tag{75}$$

with $\langle\widetilde{\xi(t)}\widetilde{\xi(t')}\rangle = \frac{2\Gamma_m k_B T}{m\omega_d^2}\delta(t-t')$ [69]. Note that we use the rotating wave approximation (RWA) to remove fast oscillating terms.

In order to remove the fast frequency terms $\cos(\omega_d t)$ and $\sin(\omega_d t)$, we additionally write the spins in the frame rotating at their Larmor frequency:

$$\widetilde{I_x} = I_x \cos(\omega_L t) - I_y \sin(\omega_L t), \tag{76}$$

$$\widetilde{I_y} = I_x \sin(\omega_L t) + I_y \cos(\omega_L t), \tag{77}$$

$$\widetilde{I_z} = I_x. \tag{78}$$

As $\omega_L$ and $\omega_d$ are close, i.e. $|\omega_L - \omega_d| \ll \omega_d$, we can now eliminate fast oscillating terms

at $\omega_{\mathrm{L}} + \omega_{\mathrm{d}}$ by means of the RWA once again, we get for the spins EOM:

$$\dot{\widetilde{I}_x} = -\frac{\widetilde{I}_x}{T_2} - \frac{\gamma}{2}\widetilde{I}_z(\widetilde{q_0} + \widetilde{\delta q_2})\left[G_y \sin((\omega_{\mathrm{d}} - \omega_{\mathrm{L}})t) + G_x \cos((\omega_{\mathrm{d}} - \omega_{\mathrm{L}})t)\right]$$
$$- \frac{\gamma}{2}\widetilde{I}_z\widetilde{\delta q_1}\left[G_y \cos((\omega_{\mathrm{d}} - \omega_{\mathrm{L}})t) - G_x \sin((\omega_{\mathrm{d}} - \omega_{\mathrm{L}})t)\right], \tag{79}$$

$$\dot{\widetilde{I}_y} = -\frac{\widetilde{I}_y}{T_2} - \frac{\gamma}{2}\widetilde{I}_z(\widetilde{q_0} + \widetilde{\delta q_2})\left[G_y \cos((\omega_{\mathrm{d}} - \omega_{\mathrm{L}})t) - G_x \sin((\omega_{\mathrm{d}} - \omega_{\mathrm{L}})t)\right]$$
$$- \frac{\gamma}{2}\widetilde{I}_z\widetilde{\delta q_1}\left[-G_y \sin((\omega_{\mathrm{d}} - \omega_{\mathrm{L}})t) - G_x \cos((\omega_{\mathrm{d}} - \omega_{\mathrm{L}})t)\right], \tag{80}$$

$$\dot{\widetilde{I}_z} = \frac{\xi(t) - I_0}{T_1} - \frac{\gamma}{2}(\widetilde{q_0} + \widetilde{\delta q_2})\left[G_x\left(\widetilde{I}_y \sin((\omega_{\mathrm{d}} - \omega_{\mathrm{L}})t) - \widetilde{I}_x \cos((\omega_{\mathrm{d}} - \omega_{\mathrm{L}})t)\right)\right.$$
$$\left. - G_y\left(\widetilde{I}_x \sin((\omega_{\mathrm{d}} - \omega_{\mathrm{L}})t) + \widetilde{I}_y \cos((\omega_{\mathrm{d}} - \omega_{\mathrm{L}})t)\right)\right]$$
$$- \frac{\gamma}{2}\widetilde{\delta q_1}\left[G_x\left(\widetilde{I}_y \cos((\omega_{\mathrm{d}} - \omega_{\mathrm{L}})t) + \widetilde{I}_x \sin((\omega_{\mathrm{d}} - \omega_{\mathrm{L}})t)\right)\right.$$
$$\left. - G_y\left(\widetilde{I}_x \cos((\omega_{\mathrm{d}} - \omega_{\mathrm{L}})t) - \widetilde{I}_y \sin((\omega_{\mathrm{d}} - \omega_{\mathrm{L}})t)\right)\right]. \tag{81}$$

with $\widetilde{q_0} = F_0/(m\omega_{\mathrm{d}}\Gamma_{\mathrm{m}})$ the rotating frame coherent drive.

Inserting the rotating frame spins in the mechanical resonator EOM gives:

$$\ddot{\widetilde{\delta q_1}} = -2\dot{\widetilde{\delta q_2}}\omega_{\mathrm{d}} - \Gamma_{\mathrm{m}}(\dot{\widetilde{\delta q_1}} + \widetilde{\delta q_2}\omega_{\mathrm{d}}) + \frac{\gamma\hbar}{m}\left[G_x\left(\widetilde{I}_x \cos((\omega_{\mathrm{d}} - \omega_{\mathrm{L}})t) - \widetilde{I}_y \sin((\omega_{\mathrm{d}} - \omega_{\mathrm{L}})t)\right)\right.$$
$$\left. + G_y\left(\widetilde{I}_y \cos((\omega_{\mathrm{d}} - \omega_{\mathrm{L}})t) + \widetilde{I}_x \sin((\omega_{\mathrm{d}} - \omega_{\mathrm{L}})t)\right)\right] + \widetilde{\xi(t)}, \tag{82}$$

$$\ddot{\widetilde{\delta q_2}} = 2\dot{\widetilde{\delta q_1}}\omega_{\mathrm{d}} - \Gamma_{\mathrm{m}}(\dot{\widetilde{\delta q_2}} - \widetilde{\delta q_1}\omega_{\mathrm{d}}) + \frac{\gamma\hbar}{m}\left[G_x\left(\widetilde{I}_x \sin((\omega_{\mathrm{d}} - \omega_{\mathrm{L}})t) + \widetilde{I}_y \cos((\omega_{\mathrm{d}} - \omega_{\mathrm{L}})t)\right)\right.$$
$$\left. + G_y\left(\widetilde{I}_y \sin((\omega_{\mathrm{d}} - \omega_{\mathrm{L}})t) - \widetilde{I}_x \cos((\omega_{\mathrm{d}} - \omega_{\mathrm{L}})t)\right)\right] + \widetilde{\xi(t)}. \tag{83}$$

We see that all fast oscillating terms have been removed. Note that the gradient in the $z$ direction ($G_z$) does not appear as we neglected its effect for the rotating frame ($G_z = 0$). We then use the same explicit Runge-Kutta method of order 8 to numerically evolve our EOM [43]. In order to extract the frequency shift from the simulated data, we calculate the instantaneous resonator phase $\phi_m$. In the rotating frame, it is given by:

$$\phi_m = \arctan\left(-\frac{\widetilde{q_0} + \widetilde{\delta q_2}}{\widetilde{\delta q_1}}\right). \tag{84}$$

We can then convert this phase to an instantaneous frequency shift with the relation:

$$\delta\omega = -\frac{\omega_0}{2Q_{\mathrm{eff}}\tan(\phi_m)}, \tag{85}$$

with $Q_{\mathrm{eff}}$ the quality factor used for the simulation. Note that the frequency shift can also be calculated using the relation:

$$\delta\omega = \frac{\omega_0}{2Q_{\mathrm{eff}}}\Delta\phi, \tag{86}$$

where $\Delta\phi = \phi_m - \phi_m^{\text{no spins}}$. The latter phase $\phi_m^{\text{no spins}}$ is the instantaneous phase of a resonator without the spin-mechanical interaction. In our numerical simulations, we additionally evolve a resonator without spin-mechanical interaction ($G_x = G_y = G_z = 0$) and use both relations to extract the frequency shift. A comparison of both methods is displayed on Fig. 6 showing negligible difference in the frequency shift estimation methods.

We can now simulate multiple trajectories by parallelizing the time evolution of the EOM. This way, we can explore different sets of parameters, mostly the detuning between the spin Larmor frequency and the mechanical resonator's frequency. A single trajectory is shown on Fig 5, it corresponds to the point with a detuning of 10 kHz on Fig. 2. Note the fluctuating frequency shift on Fig. 5(c) with a standard deviation of $\sigma_{\delta\omega} \approx 0.7$ mHz. As the fluctuating part is much bigger than the static part (i.e. the statistical polarization is much greater than the Boltzmann polarization), it is very hard to extract a precise value of the frequency shift mean without simulating extremely long times. We get the Boltzmann polarization by simulating the same parameters but turning the spin fluctuations "off".

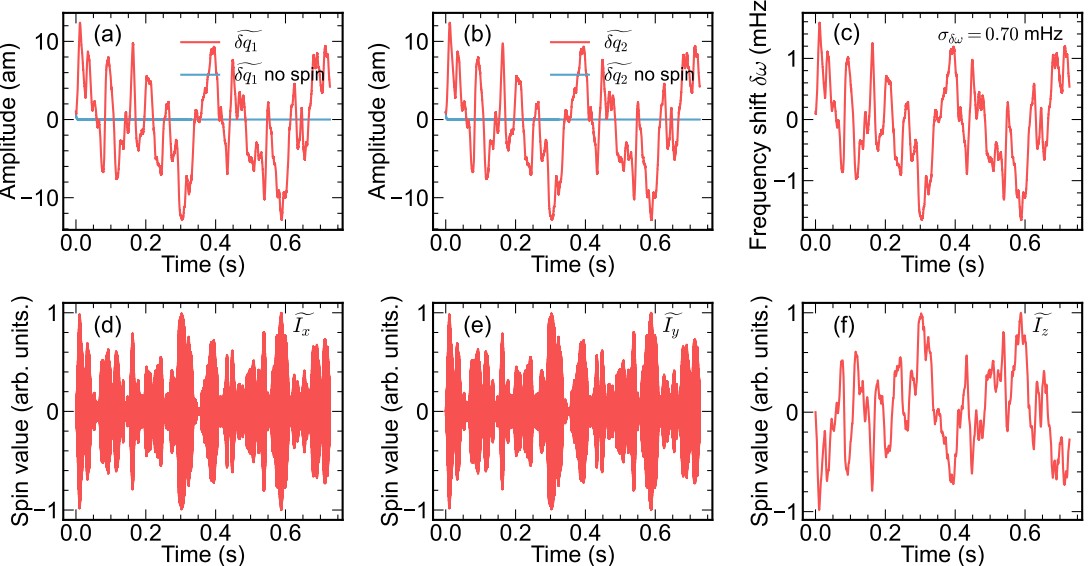

Figure 5: Single trajectory simulation for a single proton spin interacting with a mechanical resonator with frequency $\omega_0 = 2\pi \times 5.5$ MHz. The Larmor precession frequency $\omega_{\text{L}}$ of the spin is detuned by $+10$ kHz with respect to the mechanical resonator frequency. Quasi-static in-phase (a) and quadrature (b) components as defined by Eq. (73). The same components are shown for a resonator without spin-mechanical interaction. (c) Frequency shift calculated with Eq. (85). Spin components (d) $\widetilde{I_x}$, (e) $\widetilde{I_y}$ and (f) $\widetilde{I_z}$ in normalized units. Parameters are identical as Fig. 2, namely $\omega_{\text{d}} = \omega_0 = 2\pi \times 5.5$ MHz, $G_x = G_y = 6$ MT/m, $G_z = 1$ MT/m, $m = 2$ pg, $T_1 = \tau = 50$ ms, $T_2 = 100$ µs, $N = 1$ and $Q_{\text{eff}} = 2 \cdot 10^4$.

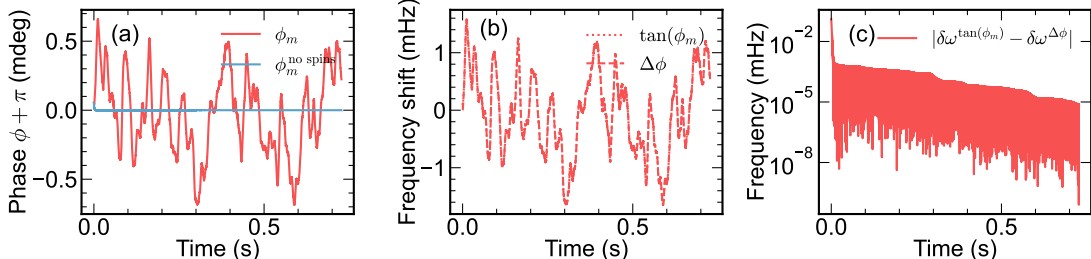

Figure 6: Phase and frequency shift of a single trajectory simulation for a single proton spin interacting with a mechanical resonator with frequency $\omega_0 = 2\pi \times 5.5\,\text{MHz}$. The Larmor precession frequency $\omega_\text{L}$ of the spin is detuned by $+10\,\text{kHz}$ with respect to the mechanical resonator frequency. (a) Phase shift of the mechanical resonator with and without spin-mechanical interaction. (b) Corresponding frequency shift using Eq.(85) and Eq. (86). (c) Comparison of (85) and Eq. (86) showing rapid convergence and negligible frequency difference. The displayed data is from the simulation showed in Fig.5.

## C.2    Magnetic tip simulations

To extract a meaningful value for the magnetic field gradients $G_i$, we perform a numerical simulation of the magnetic field of a cobalt nanomagnet. The nanomagnet resembles a cylinder of length $L = 1\,\mu\text{m}$ and radius $R = 50\,\text{nm}$. We are directly inspired by the nanomagnet presented in Ref. [70]. We assume that the nanomagnet is pre-magnetized to $1\,\text{T}$ and we apply an external magnetic field. The latter is used to tune the region where the Larmor frequency matches the mechanical resonator's frequency; we want it to be as close as possible to the nanomagnet in order to harvest the highest magnetic field gradients. Hence, the external magnetic field can be in the opposite direction of the nanomagnet $z$-magnetic field depending on the device investigated, as the required magnetic field for frequency matching can be smaller than the nanomagnet-generated magnetic field. The nanomagnet magnetization should remain roughly constant due to the shape anisotropy, which turns our Co cylinder effectively into a hard magnet [70].

Figure 7(a) shows the absolute value of the magnetic field in the vicinity of the nanomagnet (black rectangle) for the case of a SiN string with $\omega_0/2\pi = 5.5\,\text{MHz}$. In this case, we apply an external magnetic field of $0.2\,\text{T}$ in the opposite direction of the nanomagnet $z$-magnetic field. The region where the Larmor frequency of the spins would be resonant with the resonator mechanical frequency ($\omega_\text{L} = \omega_0$) is showed as a black line. We can then extract the magnetic field gradients in the $x$ and $z$ directions of the spin reference frame. These gradients are displayed on Fig. 7(b) and 7(c). In the optimal case, the sample would be in a region where $G_x$ is maximal and $G_z$ minimal. In addition, the sample must be small enough so that it does not overlap the right and left lobes otherwise the effect of the $G_x$ gradient would cancel out due to the sign inversion of the latter.

From this simulation, we extract the value of the gradients used in the main text, namely $G_x = G_y = 6\,\text{MT/m}$ ($G_x = G_y$ by symmetry) and $G_z = 1\,\text{MT/m}$.

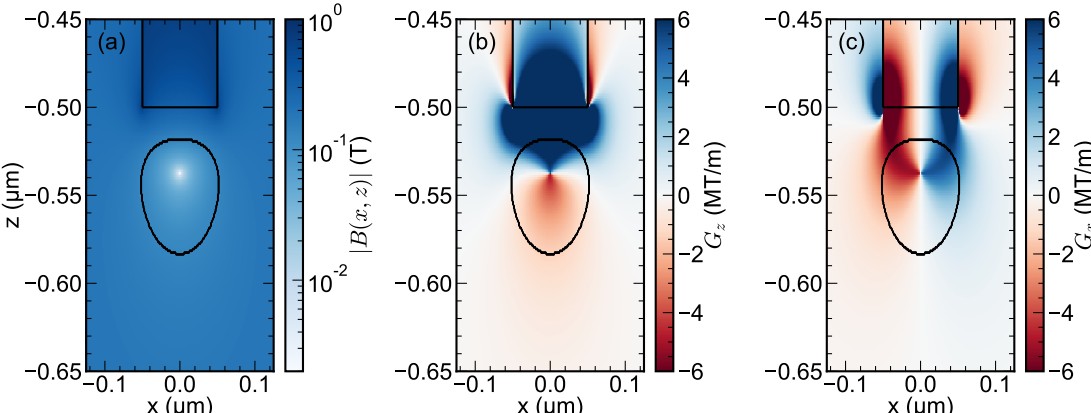

Figure 7: Numerical simulation of a cobalt nanomagnet (black rectangle) pre-magnetized at 1 T subjected to an external magnetic field of 0.2 T in the $z$-direction (bottom to top). (a) Absolute value of the magnetic field. The black line shows where the magnetic field is resonant with the mechanical resonator: $\gamma B_0 = \omega_L = \omega_0 = 2\pi \times 5.5\,\mathrm{MHz}$. The magnetic field gradients are calculated from (a) and result in a $G_z$ (b) and a $G_x$ (c) component. Note that $G_x$ and $G_z$ are the magnetic field gradients in the $x$ and $z$ directions of the spin reference frame (and not the nanomagnet reference frame).

## C.3   Boltzmann vs statistical polarization

To justify the interest in looking at the statistical polarization of the spins instead of the Boltzmann polarization, we can easily plot the different values for a range of temperature and number of spins in the sample. The Boltzmann polarization is given by Eq. (17) whereas the statistical polarization is given by $\sigma_{\delta I_0} = \frac{1}{2}\sqrt{N}$ [33]. The comparison is shown in Fig. 8 for the string resonator presented in this work. The black dashed line shows the case of $10^6$ spins. It is clear that for samples containing fewer spins the statistical polarization would allow a much stronger signal than the Boltzmann polarization in the same conditions.

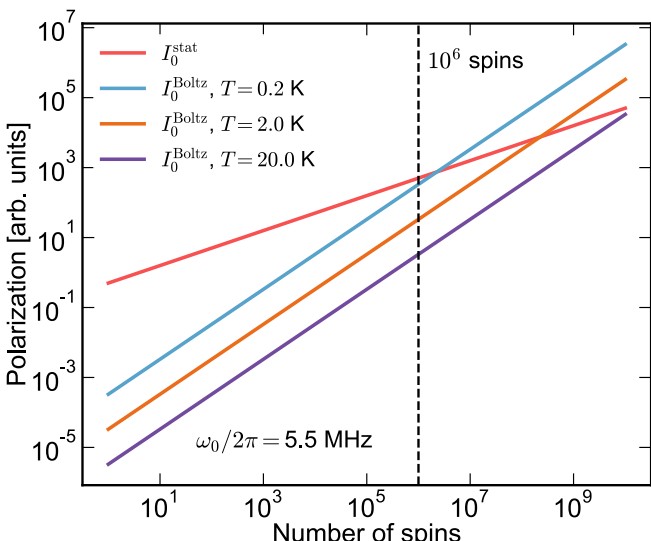

Figure 8: Boltzmann polarization compared to the statistical polarization for different numbers of spins in the sample and different temperatures for the string resonator. The black dashed line represents a sample of $10^6$ spins.

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
