# Peer review of "Near-resonant nuclear spin detection with megahertz mechanical resonators"

_SciPost Physics_

## Round 1 · Referee Report · Anonymous (Referee 2) · 2025-10-23

The referee discloses that the following generative AI tools have been used in the preparation of this report:
To read up on details of the underlying principles of spin physics, of which I am not an expert. I used ChatGPT 5.
Strengths
- Clear and rigorous theoretical framework
- Clarification of spin polarization mechanisms
- Highly relevant to nanoscale spin sensing
Weaknesses
- Frequency resolution presentation (absolute frequency instead of fractional frequency) and limited quantitative discussion of feasibility to detect such frequency fluctuations.
Report
This manuscript presents a careful and insightful theoretical study of spin detection with a nanomechanical resonator based on statistical spin polarization, which becomes significant for small spin ensembles. The authors provide a clear and rigorous theoretical framework, supported by a solid discussion and comparison with simulations. The manuscript is well written, and the methodology is transparent and logically structured.
The key insight of the paper is the fact that spin polarization arises from both thermal (Boltzmann) and statistical effects. This distinction is often overlooked, especially in standard NMR, and the authors’ discussion nicely clarifies the crossover between the thermodynamic (Boltzmann polarization dominating in macroscopic ensembles) and stochastic regimes (which dominate small spin ensembles). The analysis is directly relevant to nanoscale NMR and NV-based magnetometry, where statistical polarization can exceed the Boltzmann polarization and ultimately determine the sensitivity of spin detection.
The results are significant, and the paper will be of interest to researchers working in spin physics, magnetic resonance, and quantum sensing.
Minor comments and suggestions:
- I suggest that the authors express the frequency resolution in terms of fractional frequency (\Delta f / f_0) rather than absolute frequency where possible. This would facilitate direct comparison between systems operating at different frequencies and make the discussion of sensitivity more general.
- A short comment on potential limitations or experimental challenges in realizing the predicted regimes would further strengthen the impact of the paper.
Overall, this is a high-quality and timely contribution that combines conceptual clarity with strong physical intuition. I recommend publication after minor revision.
Recommendation
Ask for minor revision
Strengths
1- discussion of an interesting set-up 2- potentially interesting result 3- relevant for applications.
Weaknesses
1- very lengthy, yet unclear presentation 2- main point not explained in physical terms 3- unclear treatment/motivation of the stochastic terms or the bath generating the stochastic terms.
Report
for measurement of properties of the nuclear spin system.
Theoretical calculations on the level of equations of motion
are done including stochastic terms. The ambitious claim
is that even effects of a single nuclear spin can be assessed.
In my view, the result can be considered interesting if reliable,
but it is ill-presented. The main issue is illustrated in Fig. 2.
It must be justified by physical insight why the average shift
is in the range of micro Hz while the standard deviation is in
the range of milli Hz. The underlying equations are essentially
linear so that a shift due to the nuclear spin should be of the same
order of magnitude as its fluctuations - definitely in the case
of a single spin.
Since this is the crucial point of the paper it needs to be
elucidated beyond any doubt.
Wouldn't the dynamics of single spin be dominated by quantum
mechanical effects?
Note that a single spin would not fluctuate *unless* it is
coupled to some bath - which, however, does not appear in the
description.
Furthermore, there are some comments on presentation:
1)
Fig. 1 is not very clear - where is the resonator? How
would it act on the nuclear spins? Where would that be located?
Panel (b) is very schematic and again not clear at all.
In panel (c) I wonder whether the "driven amplitude" isn't
rather a "driving amplitude".
2)
Please discuss the origin of 1/T1 and 1/T2.
3)
The titles in the bibliography need to be edited for
consistency, also small and large letters.
All in all, I have some reservations about this manuscript
so that I do not recommend publication in the flagship journal
of SciPost.
Requested changes
1- The main point is to explain clearly in physical terms why the fluctuations should be easier to measure than the average shifts.
2- Fig. 1 is not very clear - where is the resonator? How would it act on the nuclear spins? Where would that be located? Panel (b) is very schematic and again not clear at all. In panel (c) I wonder whether the "driven amplitude" isn't rather a "driving amplitude".
3- Please discuss the origin of 1/T1 and 1/T2.
4- The titles in the bibliography need to be edited for consistency, also small and large letters.
Recommendation
Reject
The basic idea is to couple a mechanical resonator to nuclear spins for measurement of properties of the nuclear spin system. Theoretical calculations on the level of equations of motion are done including stochastic terms. The ambitious claim is that even effects of a single nuclear spin can be assessed.
We thank the Reviewer for their thoughtful assessment and for recognizing the interest of our results.
In my view, the result can be considered interesting if reliable, but it is ill-presented. The main issue is illustrated in Fig. 2. It must be justified by physical insight why the average shift is in the range of micro Hz while the standard deviation is in the range of milli Hz. The underlying equations are essentially linear so that a shift due to the nuclear spin should be of the same order of magnitude as its fluctuations - definitely in the case of a single spin. Since this is the crucial point of the paper it needs to be elucidated beyond any doubt.
We are happy to provide deeper physical insight into the reason why the standard deviations of spin fluctuations can exceed their average polarization. This statement is well known in the nanoscale spin community and by no means controversial, as it follows from the basic properties of the binomial distribution that can be used to model spin ensembles. Essentially, there is no fundamental reason why the width of a probability distribution should not be larger than the distribution's mean value (which can be identically zero).
Specifically for a spin ensemble, the Boltzmann (thermal) polarization $I_0 \propto N/T$ scales linearly with the number of spins $N$ and depends on temperature $T$. The polarization fluctuations stem from the statistical polarization $\sigma_{I_0} \propto \sqrt{N}$, which reflects random spin orientation and is independent of temperature. As a result, for small ensembles or a single spin, the standard deviation naturally exceeds the mean polarization by several orders of magnitude. As the resonator responds to both the mean and statistical polarization linearly, the resonator's frequency fluctuation can similarly exceed the mean frequency shift, consistent with our analytical treatment, which results in Eqs. (7) and (9). We already discussed this physics briefly after Eqs. (4-6), a discussion we now modified (see below). A detailed derivation of the underlying principle is shown in Ref. [33] of our resubmission, and experimental verifications are provided in Refs. [33] and [46]. Practical applications thereof are shown, for example, in Refs. [3-6].
Action taken: We acknowledge that this distinction was not clearly communicated in the original manuscript. The revised version now explains these points explicitly and highlights the connection between Eqs. (7) and (9).
Wouldn't the dynamics of single spin be dominated by quantum mechanical effects? Note that a single spin would not fluctuate unless it is coupled to some bath - which, however, does not appear in the description.
For the case presented in this work, the single spin dynamics is always dominated by thermal effects, as we have typical values of $k_B T$ roughly $10^3$ times larger than both $\hbar\omega_0$ and $\hbar\omega_L$. Under these conditions, the spin ensemble obeys Langevin equations with an additive stochastic term $\delta I_0(t)$, which causes pure dephasing (Eqs. (5–6) in the main text). These equations are consistent with a fluctuation–dissipation relation. In fact, this relation, along with Eqs. (5–6), can be found by coupling the ensemble to a thermally-occupied bath of harmonic oscillators, and applying time-dependent perturbation theory, as derived in standard treatments such as Gardiner and Zoller (Ref. [66] of our resubmission). The coupling of the spin to a bath is therefore already included in our description through the stochastic terms in the Langevin equations. The finite correlation time of the fluctuating spin bath $\tau$ is introduced by imposing $\delta I_0(t)$ follows an Ornstein–Uhlenbeck (OU) process, which represents a stationary, Gaussian, Markovian process. The timescale $\tau$ is system dependent and has an upper bound at the spins' decay time $T_1$. For an example of the measurement of such a correlation time, see Ref. [33] cited in our resubmission.
Action taken: We made the origin of the statistical polarization term more explicit in the text, clarifying that it arises from coupling to a fluctuating bath that induces dephasing on the spin of interest; the system is never truly isolated, even when reduced to a single spin.
Furthermore, there are some comments on presentation:
(1) Fig. 1 is not very clear - where is the resonator? How would it act on the nuclear spins? Where would that be located? Panel (b) is very schematic and again not clear at all. In panel (c) I wonder whether the "driven amplitude" isn't rather a "driving amplitude".
In Fig. 1, the resonator is shown as a wavy gray line, corresponding to a single vibrational mode seen from the side. The nuclear spins are located on top of the membrane, as illustrated in the inset. This configuration is analogous to that used in magnetic resonance force microscopy (MRFM), where a magnetic particle or a sample containing nuclear spins is positioned at the end or surface of a compliant mechanical element inside a strong magnetic field gradient. Moving the spin sample inside the gradient modulates the magnetic field as a function of time. The gradient also generates the coupling force between the nuclear magnetic moments and the mechanical motion of the resonator, enabling the detection of spin-induced forces or frequency shifts.
Panel (b) is inspired by Fig. 6 of Ref. [25] of our resubmission. This is a rather standard way of depicting different elements in a model and the corresponding coupling rates and timescales: in our case, it shows a spin (Bloch sphere) and a mechanical resonator which are coupled to each other, and which both are also coupled to a thermal bath.
Regarding the notation, by “driven amplitude” we do refer to the component of the membrane motion that results from coherent excitation by the external drive. This contribution is distinct from (i) the external driving amplitude (in units of force) and from (ii) the fluctuating motion that arises from thermal noise and is explicitly separated in our model.
Action taken: We have clarified these points in the revised figure and caption to improve the presentation.
(2) Please discuss the origin of 1/T1 and 1/T2.
The relaxation rates $1/T_1$ and $1/T_2$ refer to the longitudinal and transverse relaxation processes of the nuclear spins. In our context, $1/T_1$ arises from energy exchange between the nuclear spins and their surrounding environment, such as phonons or nearby electronic spins, while $1/T_2$ reflects dephasing due to spin–spin interactions and low-frequency magnetic field fluctuations. We note that these are heuristic parameters and that the microscopic origins of the corresponding values are usually not known in detail for nanoscale systems.
Action taken: We now clarify these points explicitly in the text.
(3) The titles in the bibliography need to be edited for consistency, also small and large letters.
We thank the Reviewer for pointing this out. We apologize for the inconsistencies in the bibliography formatting.
Action taken: The titles have now been carefully revised to ensure uniform style and consistent capitalization throughout the reference list.
All in all, I have some reservations about this manuscript so that I do not recommend publication in the flagship journal of SciPost.
We appreciate the Reviewer’s careful evaluation and constructive criticism of our work, as well as the high standards maintained by SciPost Physics. We have carefully revised the manuscript to address all points raised and to improve clarity and presentation. We have worked to strengthen the physical grounding and consistency of our approach, and we believe the revised version now conveys the reliability and relevance of the results more clearly. We are grateful for the Reviewer’s constructive feedback and sincerely hope that the revised version meets the standards of SciPost Physics and resolves all concerns.

Author: Diego Visani on 2025-11-14 [id 6035]
(in reply to Report 2 on 2025-10-23)We thank the Reviewer for the positive and encouraging comments. We are glad that the analysis and presentation were found clear and rigorous, and that the discussion of Boltzmann and statistical spin polarization was appreciated. We are grateful for the recognition of the work’s relevance to spin physics, magnetic resonance, and quantum sensing.
Action taken: We thank the Reviewer for the suggestion and now express the frequency resolution as a fractional quantity (\delta\omega/\omega_0) in the text. We link it to the additional Refs.~[48-49] of our resubmission, showcasing experimental measurement of fractional frequencies with similar mechanical devices. We chose to keep the vertical axis of Fig.~2 as is in order to avoid axis labels with many digits.
Action taken: We have added a short discussion outlining the main experimental challenges and limitations in accessing the predicted regimes.
We thank the Reviewer for the positive evaluation and recommendation for publication. We appreciate the kind remarks on the clarity and physical insight of our work.

---

## Round 2 · Referee Report · Anonymous (Referee 2) · 2025-11-21

Report

The author improved the manuscript according to my minor recommendations. I can recommend the publication in its current form.

Recommendation

Publish (easily meets expectations and criteria for this Journal; among top 50%)

---

## Round 2 · Referee Report · Anonymous (Referee 1) · 2025-12-4

Strengths

1- theoretical detailed analysis with message to experiment

Weaknesses

1- quite lengthy

Report

I thank the authors for their clarifications. I did not enough appreciate the considered high temperature limit in which the fluctuations can exceed the average polarization as discussed.
All in all, I can now agree to publication in SciPost Physics.

Still, I find Fig. 1 rather a riddle! Don't worry, I understand what you are showing. But I suggest to add another figure which presents a sketch of a (experimental) setup, e.g. with a cantilever and a driving kind of mechanism so that the reader gets an idea of the discussed apparatus. In your response, you refer to other papers; but your nice work should be self-contained.

Requested changes

see report

Recommendation

Publish (easily meets expectations and criteria for this Journal; among top 50%)

---

## Round 2 · Author Response

We hereby resubmit our manuscript entitled “Near-resonant nuclear spin detection with megahertz mechanical resonators” for further consideration in SciPost Physics.

We have thoroughly revised the manuscript in response to the detailed comments of Referee 1. In particular, we have clarified the physical origin and magnitude of polarization fluctuations relative to the mean shift, made the role of the fluctuating bath and the relaxation times T1 and T2 more explicit, and improved the presentation of Fig. 1 and the bibliography formatting. Referee 2 was very positive about the work and its relevance, and we have also implemented their suggestions.

A detailed, point-by-point reply to each Referee is provided in the separate response form, where we address all comments and indicate the corresponding changes in the revised manuscript.

We hope that the revised version meets the standards of SciPost Physics and look forward to your assessment.

Sincerely,

Diego A. Visani, Letizia Catalini, Christian L. Degen,
Alexander Eichler, and Javier del Pino

---

## Round 2 · List of Changes

1. Clarified the relation between polarization fluctuations and mean frequency shift, especially for small spin ensembles and single spins, and explicitly linked this to the main analytical results.
  2. Made the role of the fluctuating bath and the origin of the stochastic terms in the Langevin equations more explicit, including the meaning of the correlation time and its connection to relaxation processes.
  3. Added a clearer explanation of the physical origin of the relaxation rates 1/T1 and 1/T2 in nanoscale nuclear spin systems.
  4. Improved Fig. 1 and its caption to clearly indicate the position of the resonator and spins and their coupling mechanism.
  5. Revised the bibliography to ensure consistent formatting and capitalization of all titles.
  6. Adopted a fractional frequency representation for the frequency resolution where appropriate and connected it to relevant experimental work.
  7. Added a short discussion of the main experimental limitations and challenges to realizing the predicted regimes.
  8. Polished the manuscript text to improve clarity, structure, and overall presentation in the discussion of physical interpretation and relevance to nanoscale NMR and quantum sensing.

---

## Editorial Decision

in_voting